# Activity-dependent synapse elimination requires caspase-3 activation

Zhou Yu*†‡, Andrian Gutu†, Namsoo Kim, Erin K O'Shea*

Janelia Research Campus, Howard Hughes Medical Institute, Ashburn, United States

## eLife Assessment

This study presents an **important** finding on the involvement of a Caspase 3-dependent pathway in the elimination of synapses for retinogeniculate circuit refinement and eye-specific territory segregation. This work fits well with the concept of "synaptosis" which has been proposed in the past. The evidence supporting the claims of the authors is **convincing**, demonstrating that caspase-3 activation is essential for microglial elimination of synapses during both brain development and neurodegeneration. The work will be of interest to investigators studying cell death pathways, neurodevelopment, and neurodegenerative disease.

**\*For correspondence:**
zyu3@bwh.harvard.edu (ZY);
osheae@hhmi.org (EKO'S)

†These authors contributed equally to this work

**Present address:** ‡Brigham and Women's Hospital, Harvard Medical School, Boston, United States

## Abstract

During brain development, synapses are initially formed in excess and are later eliminated in an activity-dependent manner. Weak synapses are preferentially removed, but the mechanism linking neuronal activity to synapse removal is unclear. Here, we show that, in the developing mouse visual pathway, inhibiting synaptic transmission induces postsynaptic activation of caspase-3. Caspase-3 deficiency results in defects in synapse elimination driven by both spontaneous and experience-dependent neural activity. Notably, caspase-3 deficiency blocks activity-dependent synapse elimination, as evidenced by reduced engulfment of inactive synapses by microglia. Furthermore, in a mouse model of Alzheimer's disease, caspase-3 deficiency protects against synapse loss induced by amyloid-β deposition. Our results reveal caspase-3 activation as a key step in activity-dependent synapse elimination during development and synapse loss in neurodegeneration.

## Introduction

Synapse elimination is the process in which excess synapses formed early during brain development are subsequently eliminated to form mature circuits (*Kano and Hashimoto, 2009*; *Faust et al., 2021*; *Riccomagno and Kolodkin, 2015*). The outcome of synapse elimination depends on both spontaneous and experience-driven neural activity, with less active synapses being preferentially removed (*Faust et al., 2021*; *Katz and Shatz, 1996*; *Hua and Smith, 2004*). However, the molecular mechanisms that connect neuronal activity to synapse elimination remain unclear. Previous studies have proposed that microglia and astrocytes, two major glial cell types of the brain, mediate activity-dependent synapse elimination by engulfing weak synapses (*Stevens et al., 2007*; *Schafer et al., 2012*; *Li et al., 2020*; *Scott-Hewitt et al., 2020*; *Chung et al., 2013*; *Wilton et al., 2019*; *Chung et al., 2015*; *Li and Barres, 2018*; *Paolicelli et al., 2011*), but it is unclear how glia can detect the 'strength' of a synapse. A recent study proposed a molecular mechanism for activity-dependent synapse elimination in which the JAK2-STAT1 pathway is activated in inactive callosal projection neurons and regulates the removal of synapses and axons (*Yasuda et al., 2021a*). However, the JAK-STAT pathway canonically functions through transcriptional regulation (*Philips et al., 2022*), which affects entire cells. If not all synapses made by a neuron are destined for removal (*Riccomagno and Kolodkin, 2015*), activation of the JAK2-STAT1 pathway alone may not provide sufficient elimination specificity. Therefore, other

unidentified, locally effective mechanisms likely exist that confer elimination specificity at the synapse level.

Caspase-3 is a protease crucial for the execution of apoptosis (*McIlwain et al., 2013*). In hippocampal neurons, transient activation of caspase-3 is required for long-term depression (LTD), a form of synaptic plasticity that induces long-lasting decreases in synaptic strength (*Li et al., 2010*). Caspase-3 activation in apoptotic cells is also known to lead to the clearance of these cells by phagocytes (*Park and Kim, 2017*; *Trouw et al., 2008*). These findings led us to hypothesize that caspase-3 may link synapse weakening with synapse removal by glia. Although caspases were previously implicated in dendrite pruning in metamorphosis, degeneration-like axon elimination, and the regulation of hippocampal spine density (*Williams et al., 2006*; *Kuo et al., 2006*; *Simon et al., 2012*; *Ertürk et al., 2014*), the role of caspase-3 in synapse elimination in response to neural activity was not clear. In this work, we identify caspase-3 as a key molecule that is activated in postnatal development in dendritic compartments upon synaptic weakening and is necessary for the elimination of weak synapses by microglia. Furthermore, we discovered that caspase-3 deficiency protects against amyloid-β-induced synapse loss in a mouse model of Alzheimer's disease, highlighting its significance not only in developmental synapse elimination but also in adult neurodegenerative diseases.

## Results

### Tetanus toxin expression inactivates retinogeniculate synapses

We chose the mouse retinogeniculate visual pathway as a model system to study synapse elimination (*Liang and Chen, 2020*). In mice, retinal ganglion cells (RGCs) in each eye send out axons to form retinogeniculate synapses with relay neurons in the dorsal lateral geniculate nuclei (dLGN) of the thalamus on both sides of the brain (*Figure 1—figure supplement 1*; *Liang and Chen, 2020*). The majority of RGC axons from each eye innervate the contralateral (on the side opposite to the originating eye) dLGN while a smaller fraction innervates the ipsilateral (on the same side as the originating eye) dLGN (*Figure 1—figure supplement 1*; *Liang and Chen, 2020*). Regions in each dLGN innervated by the two eyes initially overlap but later segregate into eye-specific territories (*Figure 1—figure supplement 1*). This segregation process is a hallmark of retinogeniculate pathway development at the morphological level and depends on synapse elimination and spontaneous RGC activity (*Liang and Chen, 2020*; *Penn et al., 1998*; *Stellwagen and Shatz, 2002*).

To investigate the role of caspase-3 in activity-dependent synapse elimination, we first needed to establish a method that could manipulate the strength of a subset of retinogeniculate synapses. For this purpose, we used adeno-associated virus (AAV) to deliver a construct expressing tetanus toxin light chain (TeTxLC) under the control of the neuron-specific human synapsin promoter (AAV-hSyn-TeTxLC) (*Figure 1A*; *Yasuda et al., 2021b*). In utero intraocular injection of AAV-hSyn-TeTxLC at embryonic day 15 (E15) leads to TeTxLC expression in RGCs by the time of birth, blocking neurotransmitter release at retinogeniculate synapses by cleaving synaptobrevin and leading to synapse inactivation (*Yasuda et al., 2021b*; *Schiavo et al., 1992*). To label RGC axons and facilitate quantification, we co-injected AAVs that express fluorescent proteins (mTurquoise2, eGFP, or tdTomato, depending on the experiment) as anterograde tracers (*Figure 1A*). These AAVs expressing fluorescent proteins also served as controls for eye injections.

To validate the efficacy of our synapse inactivation method, we injected AAV-hSyn-TeTxLC into the right eye of wild-type E15 embryos and analyzed the segregation of eye-specific territories at postnatal day 8 (P8), when the segregation process is largely complete (*Liang and Chen, 2020*). To quantify the overlap between eye-specific territories in an unbiased manner, we applied a set of increasing thresholds to signals from both eyes (*Figure 1—figure supplement 2*; *Torborg and Feller, 2004*). For each threshold, we calculated a percentage overlap value as the ratio of the dLGN area with signals from both eyes to the total dLGN area (*Figure 1—figure supplements 2 and 3*). Consistent with previous results (*Penn et al., 1998*; *Stellwagen and Shatz, 2002*), inactivating right eye-associated retinogeniculate synapses led to contraction of right eye-specific territories and expansion of left eye-specific territories relative to control animals (*Figure 1—figure supplement 3A–B*). Importantly, segregation of eye-specific territories in TeTxLC-injected mice was significantly defective (*Figure 1—figure supplement 3C–D*), confirming that our method potently inactivated synapses and perturbed activity-dependent refinement of the retinogeniculate pathway.

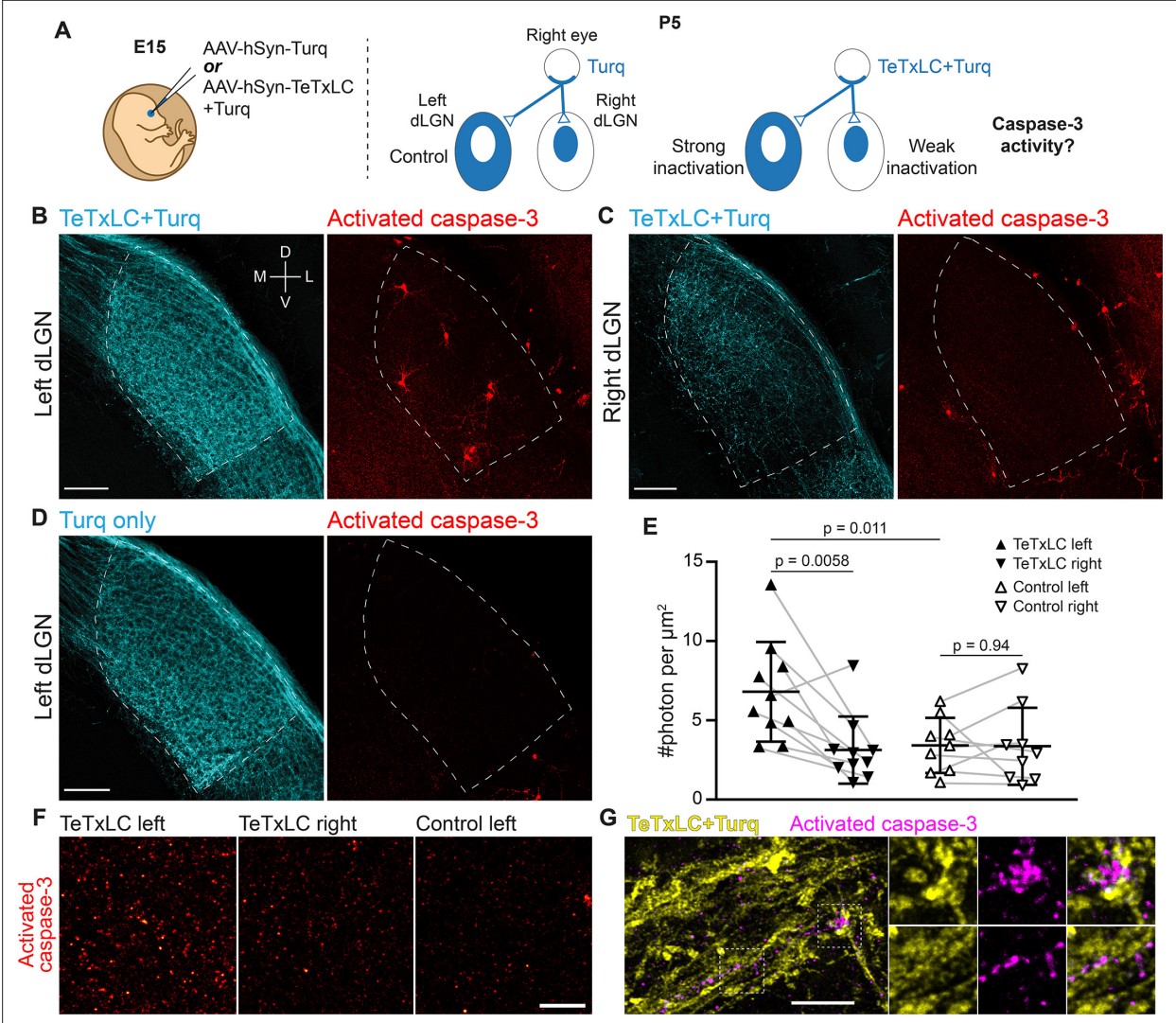

**Figure 1.** Inactivation of retinogeniculate synapses induces caspase-3 activity. (**A**) Schematics of experimental setup. Adeno-associated viruses (AAVs) expressing tetanus toxin light chain (TeTxLC) and/or mTurquoise2 (Turq) were injected into the right eye of E15 mice (left). By P5, retinogeniculate synapses in dorsal lateral geniculate nuclei (dLGN) were inactivated to varying extents depending on injection and side (right). (**B–D**) Confocal images of Turq (left panels) and activated caspase-3 (right panels) in left dLGN (**B**) and right dLGN (**C**) of a TeTxLC-injected P5 animal and in left dLGN of a control P5 animal (**D**). Dotted lines delineate dLGN boundaries. Only signals within dLGNs were analyzed. Images from the same fluorescent channel were adjusted to the same contrast. The compass in (**B**) marks tissue orientation. Scale bars: 100 μm. D, dorsal; V, ventral; M, medial; L, lateral. (**E**) Quantification of caspase-3 activity in indicated dLGNs. Activated caspase-3 signals in each dLGN (highlighted areas in **B–D**) were summed and normalized to dLGN area. Each point represents the result from one dLGN. Data from two dLGNs of the same animal were paired for analysis (gray lines). n=10 for TeTxLC-injected animals and n=9 for control animals. Mean and standard deviation (S.D.) are shown. p-values were calculated from two-tailed t-tests (paired when applicable). (**F**) Example images showing punctate caspase-3 activities in ventral-medial regions of indicated dLGNs. Images were adjusted to the same contrast. Scale bar: 20 μm. (**G**) High-resolution images of dLGN showing TeTxLC-expressing RGC axons (yellow) and activated caspase-3 (magenta). Two regions of interest (dotted squares) are magnified to illustrate that caspase-3 activity was found juxtaposing TeTxLC-expressing axon terminals but not within them. Scale bar: 5 μm.

The online version of this article includes the following figure supplement(s) for figure 1:

**Figure supplement 1.** Segregation of eye-specific territories in the mouse retinogeniculate pathway.

**Figure supplement 2.** Quantifying eye-specific segregation with multi-threshold overlap analysis.

**Figure supplement 3.** Mono-ocular blockage of retinal ganglion cell (RGC) activity with tetanus toxin light chain (TeTxLC) disrupts eye-specific segregation.

**Figure supplement 4.** Inactivation of retinogeniculate synapses induces postsynaptic caspase-3 activity in dendritic compartments of dorsal lateral geniculate nuclei (dLGN) relay neurons.

## Synapse inactivation induces postsynaptic caspase-3 activity

Is caspase-3 activated in response to synapse weakening? To address this question, we injected AAV-hSyn-TeTxLC into the right eyes of E15 embryos and looked for caspase-3 activation in dLGNs of TeTxLC-injected mice at the age of P5 (a time when synapse elimination is most active) (*Figure 1A*; *Schafer et al., 2012*; *Liang and Chen, 2020*). We used immunohistochemistry against the cleaved and active form of caspase-3 to detect caspase-3 activity in the dLGN area (*McIlwain et al., 2013*). If synapse inactivation leads to caspase-3 activation, we would expect higher levels of caspase-3 activity in dLGNs with more inactivated synapses (*Figure 1A*). In agreement with this hypothesis, caspase-3 activity in left dLGNs of TeTxLC-injected animals (dLGNs with the most inactivated synapses) was significantly higher compared to that in right dLGNs of the same animals (dLGNs with intermediate amount of inactivated synapses) (*Figure 1B, C and E*) and compared to that in dLGNs of animals receiving control injections in the right eyes (dLGNs with no inactivation) (*Figure 1B, D and E*). Reassuringly, caspase-3 activities in left and right dLGNs of control animals were comparable (*Figure 1E*), despite injections being administered only in the right eyes. This suggests that surgical manipulation and viral transduction had no measurable contribution to caspase-3 activation. These findings support the model that inactivation of retinogeniculate synapses leads to caspase-3 activation.

Does inactivation of retinogeniculate synapses activate caspase-3 pre-synaptically in RGCs or post-synaptically in dLGN relay neurons? Caspase-3 activity in dLGNs of TeTxLC-injected mice was present either as discrete, punctate signals (*Figure 1F*) or as staining of entire cells (*Figure 1B*, *Figure 1—figure supplement 4Bi-iii*). Both types of signals were more abundant in dLGNs with strong synapse inactivation than in control dLGNs (*Figure 1B, D and F*). Intriguingly, punctate caspase-3 activity was not found within TeTxLC-expressing RGC axons (*Figure 1G*). Instead, punctate caspase-3 activity closely juxtaposed inactivated axon terminals (*Figure 1G*) and co-localized with the dendritic marker MAP2 (*Figure 1—figure supplement 4A*). In the case where caspase-3 was activated in entire cells, the cells were positive for the neuronal marker NeuN (*Figure 1—figure supplement 4C*) and possessed relatively large round somas and multipolar dendritic arbors (*Figure 1—figure supplement 4Bi-iii*), all of which are characteristics of dLGN relay neurons (*Krahe et al., 2011*; *Parnavelas et al., 1977*). These observations suggest that caspase-3 activation triggered by synapse inactivation occurs predominantly in the dendritic compartments of dLGN relay neurons that are postsynaptic to the inactivated synapses.

Although morphologically distinct, localized and whole-cell caspase-3 activity may be mechanistically linked. In support of this, we observed transitional states of caspase-3 activation where multiple dendrites of a relay neuron were positive for active caspase-3 without the cell body becoming apoptotic (an example is shown in *Figure 1—figure supplement 4Biv*), suggesting that whole-cell caspase-3 activation may arise from accumulation of local caspase-3 activity. Such transitions might be particularly relevant in dLGNs that receive the majority of TeTxLC-expressing axons, as systematic silencing of synapses could lead to widespread caspase-3 activation in the postsynaptic neuron, potentially overwhelming negative regulatory mechanisms that normally keep caspase-3 activation localized (*Ertürk et al., 2014*; *Choi et al., 2009*; *Suzuki et al., 2001*; *McComb et al., 2019*). While our data cannot determine which form of caspase-3 activation plays a more significant role in the normal development of the visual pathway, the infrequent occurrence of whole-cell caspase-3 activation in control dLGNs (*Figure 1D*) -- which undergo normal synapse elimination -- suggests that localized caspase-3 activation is necessary to drive synapse elimination under physiological conditions.

## Caspase-3 activation at weak synapses requires the presence of strong synapses

Previous experiments have manipulated RGC neuronal activities and demonstrated that, regardless of the absolute level of RGC activity, the dLGN territory of less active RGCs always contracted relative to that of more active RGCs (*Penn et al., 1998*; *Stellwagen and Shatz, 2002*). This observation suggests that the outcome of synapse elimination is driven by competitive interactions between strong and weak synapses. Consistent with this idea, a recent study demonstrated that elimination of weak synapses in callosal projections occurred only when other strong synapses are present (*Yasuda et al., 2021b*; *Nagappan-Chettiar et al., 2023*).

To test if caspase-3 activation at weak synapses requires the presence of strong synapses, we designed two experimental conditions. In the first condition, we inactivated RGCs only in the right eye,

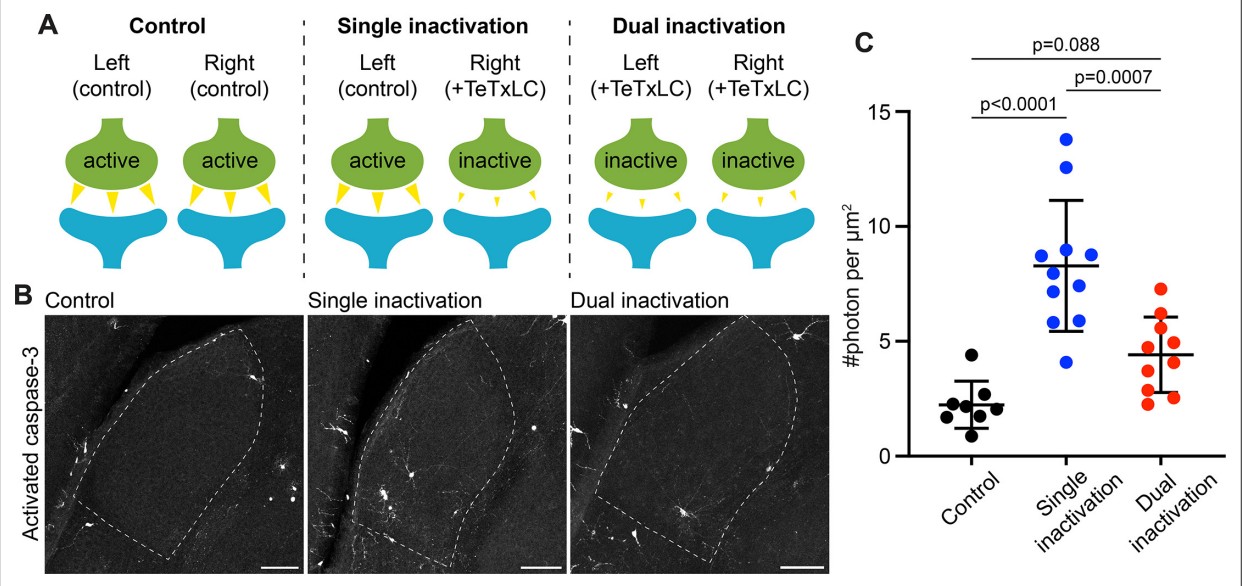

**Figure 2.** Caspase-3 activation at weak synapses requires the presence of strong synapses. (**A**) Schematics illustrating experimental conditions. No synapses are inactivated in the control (left); only synapses from right eyes are inactivated in single inactivation condition (middle); synapses from both right and left eyes are inactivated in dual inactivation condition (right). (**B**) Confocal images of P5 left side dorsal lateral geniculate nucleus (dLGNs) in the three conditions showing caspase-3 activity in the dLGN. Dotted lines mark dLGN boundaries. Scale bars: 100 μm. (**C**) Quantification of dLGN caspase-3 activity in the indicated conditions. Caspase-3 activity in each dLGN were summed and normalized to the dLGN area. For the single inactivation condition, values were from left dLGNs only. For the other two conditions, values from both dLGNs were averaged. n=8 animals for control, n=11 animals for single inactivation, and n=10 animals for dual inactivation. Mean and S.D. are shown. P-values were calculated from Tukey's multiple comparison tests.

The online version of this article includes the following figure supplement(s) for figure 2:

**Figure supplement 1.** Bi-ocular blockage of retinal ganglion cell (RGC) activity with tetanus toxin light chain (TeTxLC) disrupts eye-specific segregation.

leading to the presence of both active (from the left eye) and inactive (from the right eye) synapses in the dLGN (*Figure 2A*). We termed this condition 'single inactivation.' In the second condition, we inactivated RGCs in both eyes, leading to the presence of only inactive synapses in the dLGN (*Figure 2A*). We termed this condition 'dual inactivation.' We also included a control condition where no RGCs were inactivated (*Figure 2A*). We first confirmed that, in accordance with previous findings (*Penn et al., 1998*; *Rossi et al., 2001*), segregation of eye-specific territories was disrupted in the dual inactivation condition (*Figure 2—figure supplement 1A–D*). We then measured dLGN caspase-3 activity in control, single, and dual inactivation conditions. Intriguingly, while dLGN caspase-3 activity was significantly higher in the single inactivation condition compared to that in the control (*Figure 2B and C*), dLGN caspase-3 activity in the dual inactivation condition was not significantly different from that in the control (*Figure 2B and C*) and was significantly lower than that in the single inactivation condition (*Figure 2B and C*). Although not reaching statistical significance, dLGN caspase-3 activity on average was still higher in the dual inactivation condition compared to the control (*Figure 2C*), possibly due to incomplete inactivation of retinogeniculate synapses, or interaction between weak retinogeniculate synapses and strong synapses from other pathways, or additional mechanisms that activate caspase-3 independently of interactions between weak and strong synapses. These results suggest that interaction between weak and strong synapses is required for caspase-3 activation to occur at weak synapses.

## Caspase-3 is required for segregation of eye-specific territories

Is synapse inactivation-induced caspase-3 activity important for synapse elimination in the retinogeniculate pathway? To address this question, we investigated whether segregation of eye-specific dLGN territories is defective if caspase-3 activation does not occur. Since we were unable to find a Cre-expressing mouse line that satisfactorily depletes caspase-3 in dLGN relay neurons, we used full-body caspase-3 deficient (*Casp3$^{-/-}$*) mice (*Kuida et al., 1996*; *Houde et al., 2004*). These mice were

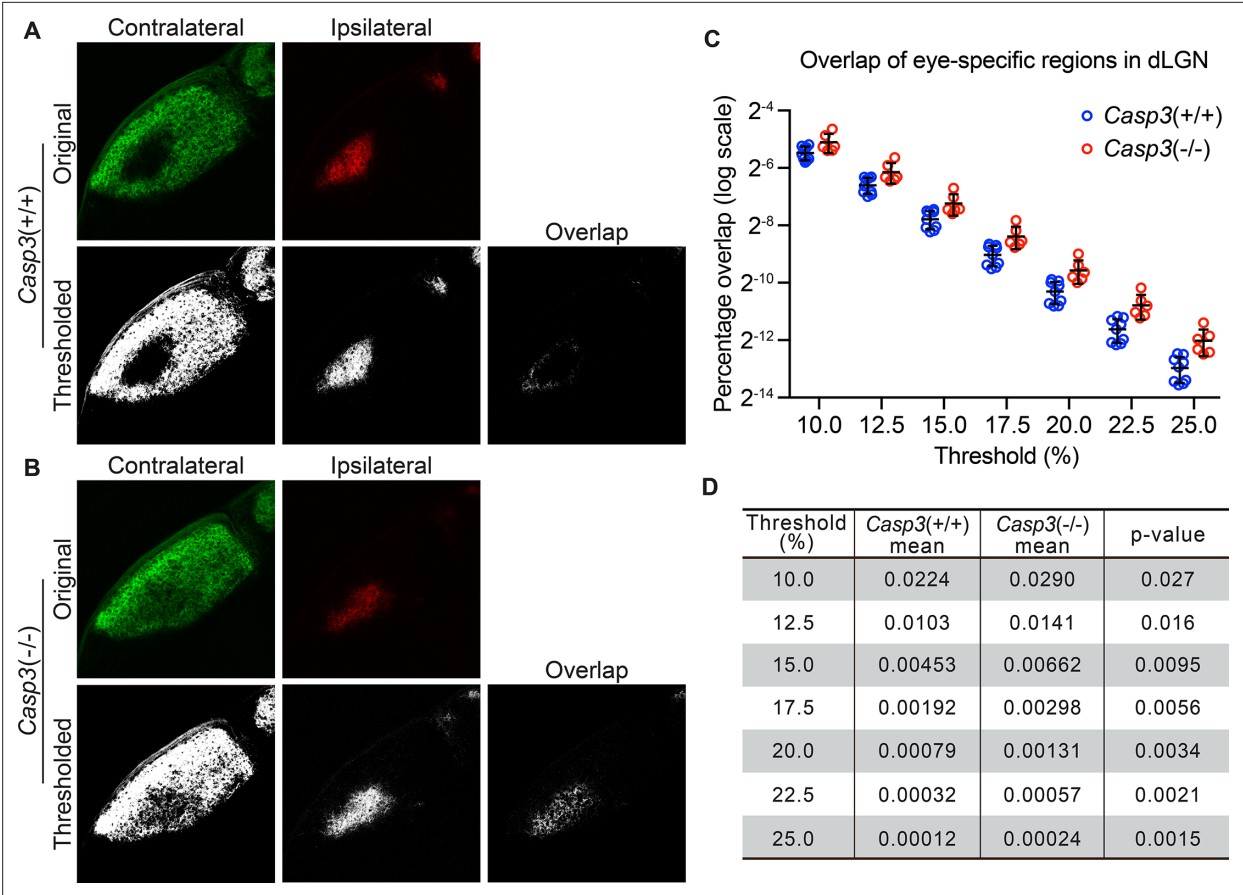

**Figure 3.** Caspase-3 is required for segregation of eye-specific territories. (**A–B**) Representative confocal images of retinogeniculate inputs in the dorsal lateral geniculate nuclei (dLGN) of P10 wild-type (**A**) and *Casp3⁻/⁻* (**B**) mice. Contralateral inputs are labeled with AlexaFluor488 (AF488) conjugated cholera toxin subunit B (CTB) and ipsilateral inputs with AF594-CTB. Original images were thresholded into 0-or-1 images using the Otsu method (*Suzuki et al., 2001*), and the overlap between thresholded contralateral and ipsilateral inputs is shown. (**C**) Percentage overlap between eye-specific territories in wild-type and *Casp3⁻/⁻* mice under a series of increasing signal cutoff thresholds. Note that the percentage overlap is plotted on a log scale. Each circle represents one animal. Mean and S.D. are shown. n=9 for wild-type mice and n=6 for *Casp3⁻/⁻* mice. (**D**) Mean percentage overlap values in wild-type and *Casp3⁻/⁻* mice and p-values of two-tailed t-tests between the two genotypes are listed for each cutoff threshold.

The online version of this article includes the following figure supplement(s) for figure 3:

**Figure supplement 1.** Caspase-3 deficiency does not alter retinal ganglion cell (RGC) density in the retina.

**Figure supplement 2.** Caspase-3 deficiency does not alter relay neuron density in the dorsal lateral geniculate nuclei (dLGN).

backcrossed to congenicity on the C57Bl/6 J background and were viable, fertile, and did not exhibit overt abnormalities (*Houde et al., 2004*).

To measure segregation of eye-specific territories, we injected the eyes of *Casp3⁺/⁺* and *Casp3⁻/⁻* littermate mice at P9 with fluorophore-conjugated cholera toxin subunit B (CTB), using a different fluorophore for each eye. CTB molecules bind to ganglioside molecules on RGC surfaces and anterogradely label RGC axon terminals (*De Haan and Hirst, 2004*). At P10, when CTB labeling and eye-specific segregation were complete (*Liang and Chen, 2020*; *Torborg and Feller, 2004*), we collected the brains and imaged eye-specific territories in each dLGN. By visual inspection, eye-specific territories in *Casp3⁺/⁺* dLGNs overlapped minimally (*Figure 3A*). In contrast, *Casp3⁻/⁻* dLGNs showed clear defects in the separation of eye-specific territories (*Figure 3B*; *Otsu, 1979*). We then quantified eye-specific segregation as described previously (*Figure 1—figure supplement 2*) and found that percentage overlap in *Casp3⁻/⁻* dLGNs was consistently higher than that in *Casp3⁺/⁺* dLGNs across all thresholds (*Figure 3C and D*). Importantly, the fold difference and statistical significance of the difference in percentage overlap between the two groups of animals increased as the cutoff threshold

increased (*Figure 3D*), suggesting the defects observed in *Casp3*[-/-] animals cannot be attributed to artifacts generated from background noise.

As RGCs are known to undergo extensive apoptosis during the first week of postnatal development (*Young, 1984*; *Dreher et al., 1983*), we wondered if caspase-3 deficiency might block RGC turnover and result in more RGC inputs in the dLGN, thereby confounding the overlap analysis. To address this concern, we quantified the density of RGCs in whole-mount retinae using an RGC marker, RBPMS (*Figure 3—figure supplement 1A*; *Rodriguez et al., 2014*). Reassuringly, RGC densities in retinae of *Casp3*[+/+] and *Casp3*[-/-] mice were not different (*Figure 3—figure supplement 1B*), indicating that RGC apoptosis still occurred at normal rates in the absence of caspase-3, presumably through redundant mechanisms.

It is also possible that caspase-3 deficiency might confound the overlap analysis by affecting the number of dLGN relay neurons. Therefore, we used NeuN staining to visualize relay neurons in dLGNs of *Casp3*[+/+] and *Casp3*[-/-] mice and confirmed that relay neuron density is not significantly altered by caspase-3 deficiency (*Figure 3—figure supplement 2A–B*). Notably, this observation indicates that, during normal development, synapse weakening does not induce widespread whole-cell caspase-3 activation in dLGN relay neurons, corroborating the idea that localized caspase-3 activity drives synapse elimination in dLGN. Collectively, our findings suggest that caspase-3 is required for normal segregation of eye-specific territories in dLGN.

## Caspase-3 is required for retinogeniculate circuit refinement

Segregation of eye-specific territories in the mouse dLGN occurs prior to eye opening (which happens at ~P13) and relies predominantly on spontaneous RGC activity (*Liang and Chen, 2020*). After eye opening, sensory-dependent RGC activity drives a second phase of refinement at the circuit level (*Liang and Chen, 2020*). At the time of eye opening, each dLGN relay neuron receives numerous weak RGC inputs. With visual experience, a minority of these inputs are strengthened while other weak inputs are eliminated, so that by P30 only a few (typically ≤3) strong RGC inputs innervate each dLGN relay neuron (*Liang and Chen, 2020*).

To test if caspase-3 is required for visual experience-driven circuit refinement, we prepared acute brain slices from P30 *Casp3*[+/+] and *Casp3*[-/-] littermate mice and investigated the electrophysiological properties of retinogeniculate synapses (*Figure 4—figure supplement 1A*; *Chen and Regehr, 2000*; *Hooks and Chen, 2006*). We stimulated the optic tract while recording both AMPA receptor (AMPAR)- and NMDA receptor (NMDAR)-mediated excitatory postsynaptic currents (EPSCs) in dLGN relay neurons (*Figure 4—figure supplement 1A*, *Figure 4A-B*). By gradually increasing the stimulation intensity on the optic tract, we recruit individual RGC axons in a serial manner (*Figure 4—figure supplement 1A*). Excitation of each RGC axon triggers a step increase in the EPSC amplitude (*Figure 4—figure supplement 1A*), allowing us to infer RGC input numbers by counting steps in EPSC response curves (*Figure 4C*, *Figure 4—figure supplement 2A*). In agreement with previous studies, retinogeniculate circuits in *Casp3*[+/+] animals were refined, with dLGN relay neurons most frequently receiving 2 strong RGC inputs, and the great majority of neurons receiving no more than 3 inputs (*Figure 4A,C*, *Figure 4—figure supplement 2A*). In contrast, RGC input counts of dLGN relay neurons in *Casp3*[-/-] animals followed a long-tailed distribution that was significantly right-shifted from that of *Casp3*[+/+] animals (*Figure 4C*, *Figure 4—figure supplement 2A*). The majority of *Casp3*[-/-] relay neurons received more than 3 inputs (*Figure 4B-C*, *Figure 4—figure supplement 2A*), and many inputs elicited only small increments in EPSC amplitude (*Figure 4B*). Additionally, we examined maximum EPSC amplitude (*Figure 4—figure supplement 2B–C*), AMPAR-EPSC to NMDAR-EPSC ratio (*Figure 4—figure supplement 2D*), and fiber fraction (*Figure 4—figure supplement 2E*; *Hooks and Chen, 2006*) in *Casp3*[+/+] and *Casp3*[-/-] mice. We found no significant difference between the two groups of animals, suggesting that the defective circuit refinement in *Casp3*[-/-] animals was not due to the lack of retinogeniculate synapse maturation but instead an inability to eliminate weak synapses.

An increased number of RGC inputs innervating dLGN relay neurons in *Casp3*[-/-] mice should correspond to greater numbers of presynaptic release sites and should result in more frequent spontaneous release of neurotransmitters. To test this prediction, we measured miniature EPSCs (mEPSCs) in dLGN relay neurons of *Casp3*[+/+] and *Casp3*[-/-] mice (*Figure 4D*). We found that mEPSCs in *Casp3*[-/-] neurons occurred more frequently (*Figure 4E*), with a greater fraction of them having small amplitudes (*Figure 4F*). These results mirror previous measurements in hippocampal CA1 neurons in

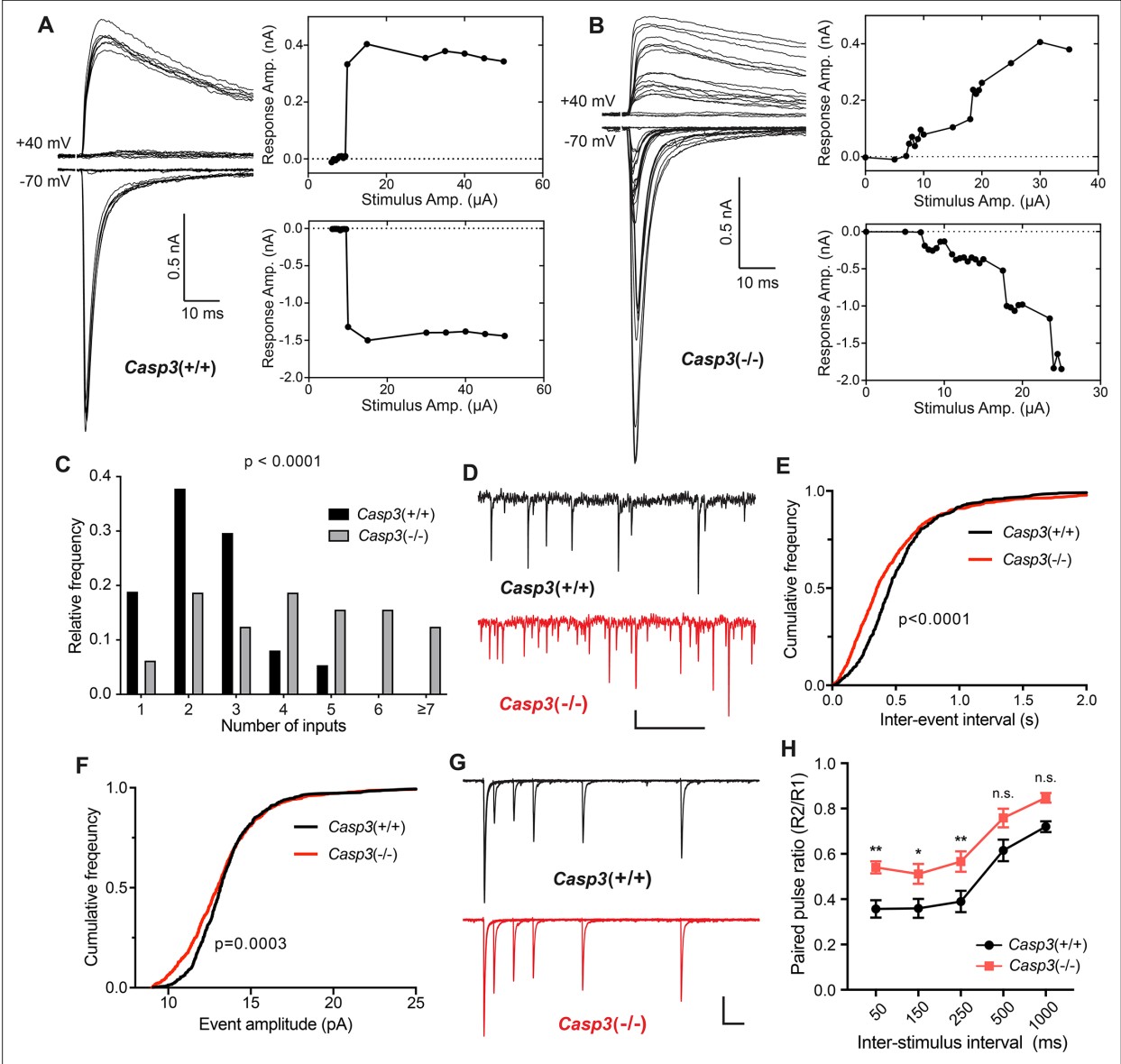

**Figure 4.** Caspase 3 is required for retinogeniculate circuit refinement. (**A–B**) Example recordings of dorsal lateral geniculate nuclei (dLGN) relay neuron responses in P30 wild-type (**A**) and *Casp3⁻ᐟ⁻* (**B**) mice. Excitatory postsynaptic currents (EPSCs) were evoked by increasing stimulation currents in the optic tract. Both AMPAR-mediated inward current at –70 mV membrane potential and AMPAR and NMDAR-mediated outward current at +40 mV membrane potential are shown. Peak response amplitudes at each stimulation intensity are plotted to the right of recording traces. Scale bars represent 0.5 nA and 10 ms. (**C**) Distribution of retinal ganglion cell (RGC) input numbers on individual dLGN relay neurons in wild-type and *Casp3⁻ᐟ⁻* mice. The number of RGC inputs was inferred by manually counting the number of steps in AMPAR-mediated EPSC response curves (lower right in **A** and **B**) while blind to the genotypes. p-value was calculated from two-tailed t-test. n=37 cells from 22 wild-type mice and n=32 cells from 16 *Casp3⁻ᐟ⁻* mice. (**D**) Example recordings of mEPSC measurements from wild-type and *Casp3⁻ᐟ⁻* mice. Scale bars represent 0.5 s (horizontal) and 5 pA (vertical). (**E–F**) Cumulative distribution curves of inter-mEPSC intervals (**E**) and mEPSC amplitudes (**F**) in wild-type and *Casp3⁻ᐟ⁻* mice. p-values were calculated from Kolmogorov-Smirnov tests. n=16 cells from 4 wild-type mice and n=17 cells from 4 *Casp3⁻ᐟ⁻* mice. (**G**) Example recordings from paired-pulse measurements at –70 mV membrane potential in wild-type and *Casp3⁻ᐟ⁻* mice. Traces from experiments with 50, 150, 250, 500, and 1000 ms inter-stimulus intervals are overlayed. Stimulus artifacts were removed from the traces for clarity. Scale bars represent 100 ms (horizontal) and 0.3 nA (vertical). (**H**) Paired-pulse ratio (calculated as amplitude of the second response over that of the first response) in wild-type and *Casp3⁻ᐟ⁻* mice at various inter-stimulus intervals. Mean and standard error of the mean (SEM) are shown. p-values were calculated from Bonferroni's multiple comparison test. p=0.0067 for 50 ms interval, p=0.0369 for 150 ms interval, and p=0.0097 for 250 ms interval. n=16 cells from seven wild-type mice and n=13 cells from 4 *Casp3⁻ᐟ⁻* mice.

The online version of this article includes the following figure supplement(s) for figure 4:

**Figure supplement 1.** Measuring electrophysiological properties of retinogeniculate synapses.

**Figure supplement 2.** Additional analyses of electrophysiological properties of retinogeniculate synapses in *Casp3⁺ᐟ⁺* and *Casp3⁻ᐟ⁻* mice.

Casp3[-/-] mice (**Ertürk et al., 2014**) and are consistent with the model where caspase-3 deficiency impairs the removal of weak synapses, leading to greater number of inputs on dLGN relay neurons.

As an additional way to probe retinogeniculate circuit refinement, we measured paired-pulse response ratios (PPRs) in dLGN relay neurons (**Figure 4—figure supplement 1B**, **Figure 4G-H**; **Chen and Regehr, 2000**; **Regehr, 2012**). The retinogeniculate circuit typically generates PPRs that are smaller than one due to depletion of neurotransmitters (**Figure 4—figure supplement 1B**; **Chen and Regehr, 2000**; **Regehr, 2012**). If there is an increase in the number of release sites in Casp3[-/-] mice, we expect more available neurotransmitters and larger PPRs (**Figure 4—figure supplement 1C**; **Regehr, 2012**). Indeed, we observed higher PPRs in Casp3[-/-] dLGN relay neurons (**Figure 4G–H**), suggesting that defective synapse elimination resulted in more RGC release sites in Casp3[-/-] mice. Taken together, our results demonstrate that caspase-3 is required for experience-driven retinogeniculate circuit refinement.

## Caspase-3 deficiency impairs synapse elimination by microglia

Our results demonstrate that proper synapse elimination in the retinogeniculate pathway requires caspase-3 activity. Previous studies have shown that microglia and astrocytes engulf weak synapses

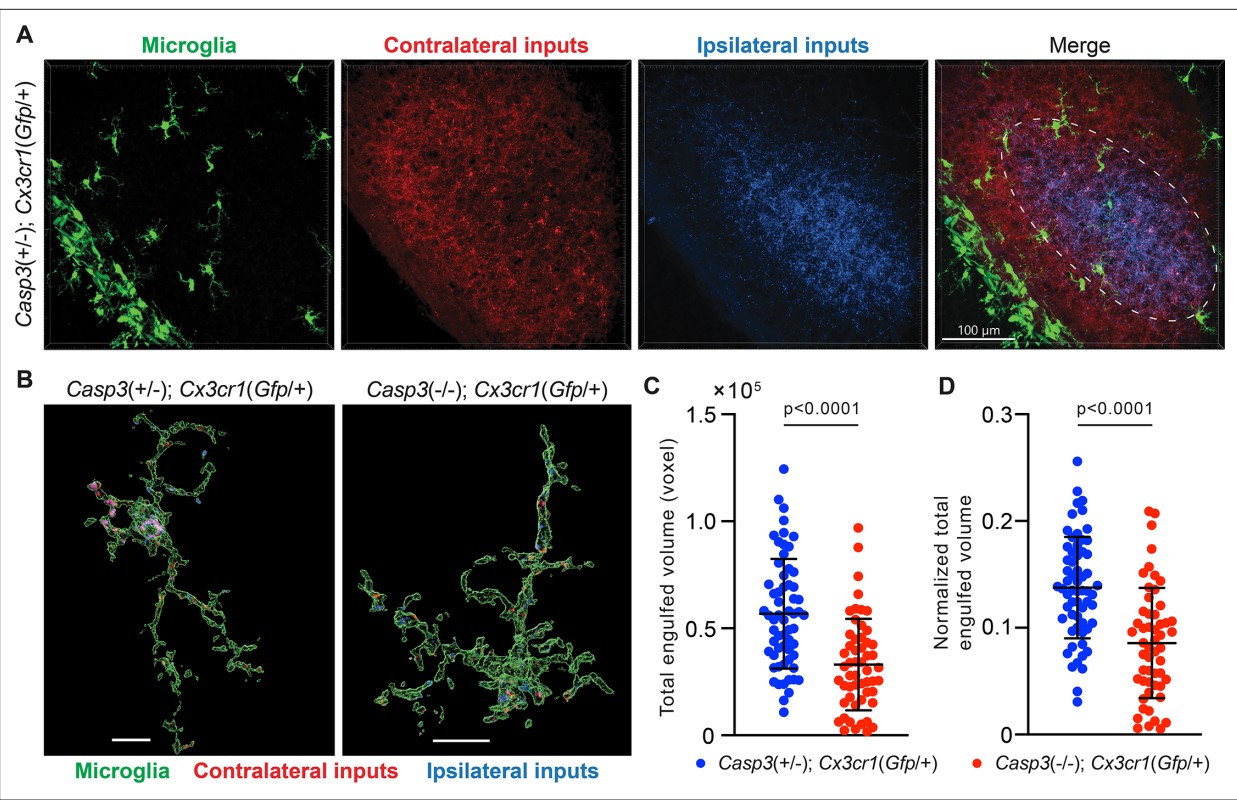

**Figure 5.** Microglia-mediated synapse elimination depends on caspase-3. (**A**) Representative 3D-reconstructed images of a P5 Casp3[+/-]; Cx3cr1[Gfp/+] mouse dorsal lateral geniculate nuclei (dLGN) with microglia displayed in green, contralateral retinal ganglion cell (RGC) axon terminals in red, and ipsilateral RGC terminals in blue. In the merged image, the region from which microglia are selected for analysis is indicated with the dashed line. All microglia within this region were analyzed. The scale bar represents 100 µm. (**B**) Representative surface rendering of microglia (green) from P5 dLGNs of Casp3[+/-]; Cx3cr1[Gfp/+] and Casp3[-/-]; Cx3cr1[Gfp/+] mice. Intracellular contralateral (red) and ipsilateral (blue) RGC axon terminals are shown. Microglia from caspase-3 deficient mice engulf visibly less synaptic material. Scale bars represent 10 µm. (**C**) Total volume of engulfed synaptic material in individual microglia from Casp3[+/-]; Cx3cr1[Gfp/+] and Casp3[-/-]; Cx3cr1[Gfp/+] mice. (**D**) Total volume of engulfed synaptic material in each microglia (from **C**) is normalized to the volume of that microglia. In (**C–D**), each point represents one microglia. Mean and S.D. are shown. p-values were calculated from unpaired two-tailed t-tests. n=61 microglia from 8 Casp3[+/-]; Cx3cr1[Gfp/+] mice and n=54 microglia from 5 Casp3[-/-]; Cx3cr1[Gfp/+] mice.

The online version of this article includes the following figure supplement(s) for figure 5:

**Figure supplement 1.** Caspase-3 deficiency does not cause microglia activation.

**Figure supplement 2.** Additional analyses of microglia-mediated engulfment of synaptic material.

**Figure supplement 3.** Astrocyte-mediated synapse elimination does not appear to depend on caspase-3.

during the process of synapse elimination (*Schafer et al., 2012*; *Chung et al., 2013*). We, therefore, sought to determine whether caspase-3 activation is required for synapse elimination by microglia and/or astrocytes.

If synapse elimination by microglia depends on caspase-3 activation, then microglia in *Casp3*⁻ᐟ⁻ mice should engulf less synaptic material. Similar to a previous study (*Schafer et al., 2012*), we utilized the *Cx3cr1*^Gfp transgenic mouse line to visualize microglia in vivo. In this line, the native *Cx3cr1* promoter drives *Gfp* expression and specifically labels microglia in the brain (*Figure 5A*; *Jung et al., 2000*). We labeled RGC axon terminals by injecting fluorophore-conjugated CTB in the eyes of P4 *Casp3*^+/-; *Cx3cr1*^Gfp/+ and *Casp3*^-/-; *Cx3cr1*^Gfp/+ littermate pups (*Casp3*^+/- instead of *Casp3*^+/+ mice were used to enable more efficient breeding) (*Schafer et al., 2012*). At P5, the brains were harvested and the dLGN was imaged (*Figure 5A*). Microglia in *Casp3*^-/-; *Cx3cr1*^Gfp/+ mice had small soma and thin processes, indicating that caspase-3 deficiency did not result in microglial activation (*Figure 5B*). This is further validated by immunohistochemical analysis of microglia in caspase-3 deficient mice (*Figure 5—figure supplement 1A–D*). To quantify synapse engulfment, we isolated volumes corresponding to microglia and RGC axon terminals in the dLGN region where both contralateral and ipsilateral inputs are present (*Figure 5A and B*) and calculated the volume of RGC terminals that were fully enclosed in every microglia within the region of interest (*Figure 5C*). As expected, we observed that microglia in *Casp3*^-/-; *Cx3cr1*^Gfp/+ mice engulfed significantly less axonal material compared to those of *Casp3*^+/-; *Cx3cr1*^Gfp/+ littermates (*Figure 5B–C*). The same trend was observed when RGC axon terminals originating from the two eyes were analyzed separately (*Figure 5—figure supplement 2A–B*). As the engulfment capacity of microglia is limited by their size, we normalized the volume of engulfed synaptic material in each microglia to the microglial volume (*Figure 5D*). Microglia in both groups of mice are comparable in size (*Figure 5—figure supplement 2C*), and the defective synaptic engulfment in *Casp3*^-/- mice persisted after normalization to microglial volume (*Figure 5D*). Additionally, the same defect was observed when engulfment values were averaged and reported by animal (*Figure 5—figure supplement 2D*). Taken together, our results demonstrate that microglia-mediated synapse elimination depends on caspase-3.

To test whether caspase-3 activation is also required for astrocyte-mediated synapse elimination, we used the *Aldh1l1-Gfp* transgenic mouse line, in which an *Aldh1l1-Gfp* cassette is inserted into the genome via BAC and specifically labels astrocytes with GFP expression (*Figure 5—figure supplement 3A*; *Gong et al., 2003*). We again labeled RGC axon terminals in *Casp3*^+/-; *Aldh1l1-Gfp*^+/- and *Casp3*^-/-; *Aldh1l1-Gfp*^+/- littermate mice with CTB conjugates (*Figure 5—figure supplement 3A*). Astrocytes were present in the P5 dLGN at a much higher density than microglia and possess very fine processes (*Figure 5—figure supplement 3A*). To ensure segmentation fidelity, we only analyzed astrocyte cell bodies and their proximal processes (*Figure 5—figure supplement 3B*). Unexpectedly, we found that the volume of axonal material engulfed by astrocytes was comparable in the two groups of mice (*Figure 5—figure supplement 3C*). The same trend was observed when we normalized the volume of engulfed synaptic material to astrocyte volume (*Figure 5—figure supplement 3D–E*). It is possible that, by using *Casp3*^+/- mice instead of *Casp3*^+/+ mice as controls, we have obscured some defects in astrocyte-mediated synapse elimination. Nevertheless, given that astrocyte-mediated synapse elimination did not show a clear dependence on caspase-3, we focused on measuring synapse engulfment by microglia as a readout of preferential elimination of weak synapses.

## Removal of weak synapses by microglia requires caspase-3 activity

If caspase-3 activation at weak synapses is important for synapse removal, then in caspase-3 deficient mice, microglia should lose the ability to engulf weak synapses. To test this hypothesis, we inactivated retinogeniculate synapses originating from right eyes in *Casp3*^+/+ and *Casp3*^-/- mice with AAV-hSyn-TeTxLC injections (*Figure 6A*). A group of animals was included for each genotype that received control injections. To measure synapse engulfment by microglia, we labeled RGC axon terminals with CTB conjugates and visualized microglia by immunostaining against Iba1, a marker specific for microglia in the brain (*Figure 6B–G*). According to our model, in TeTxLC-injected *Casp3*^+/+ mice, microglia should preferentially engulf inactive retinogeniculate synapses originating from the right eye (*Figure 6A*), whereas, in TeTxLC-injected *Casp3*^-/- mice, synapses from the right and the left eye should be engulfed at similar levels regardless of their strength (*Figure 6A*).

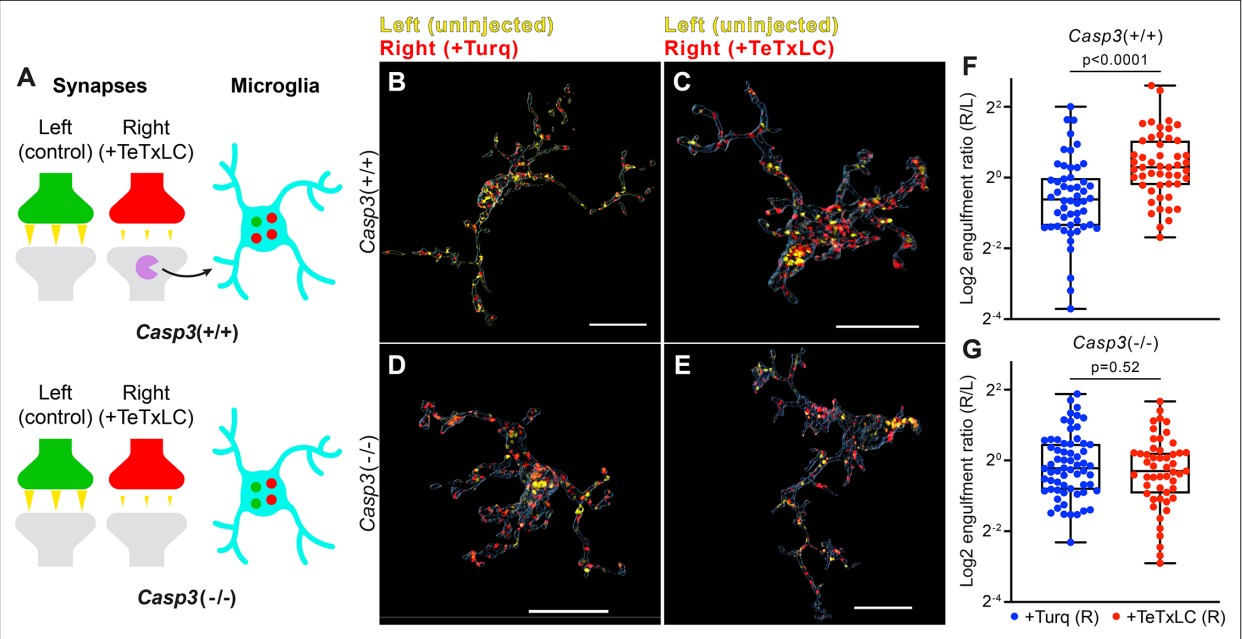

**Figure 6.** Removal of weak synapses by microglia requires caspase-3 activity. (**A**) Schematics illustrating the experimental rationale. In wild-type mice (upper panel), inactivating retinogeniculate synapses from the right eye activates caspase-3 (magenta) and recruits microglia to preferentially engulf right eye-originated synapses (red) over left eye-originated synapses (green). If caspase-3 activation is blocked (lower panel), engulfment of inactive synapses should be attenuated. (**B–E**) Surface rendering of representative microglia from P5 left dorsal lateral geniculate nuclei (dLGN) of $Casp3^{+/+}$ (**B–C**) or $Casp3^{-/-}$ (**D–E**) mice injected with adeno-associated virus (AAV) that provided mTurquoise2 (**B and D**) or tetanus toxin light chain (TeTxLC) (**C and E**) in the right eye at E15. Microglia were labeled by immunostaining against Iba1. Retinal ganglion cell (RGC) axon terminals from the left eye are shown in yellow and terminals from the right eye in red. Scale bars represent 15 μm. (**F–G**) Ratio between volumes of right-eye and left-eye-originated synaptic material engulfed by microglia from $Casp3^{+/+}$ (**F**) or $Casp3^{-/-}$ (**G**) mice injected with AAV that carried the gene for mTurquoise2 (blue) or TeTxLC (red). Each dot represents one microglia. Engulfment ratios are displayed on a log scale. 0, 25, 50, 75, and 100 percentiles are shown. p-values were calculated from unpaired two-tailed Mann-Whitney tests. n=52 microglia from 7 Turq-injected $Casp3^{+/+}$ mice, n=50 microglia from 8 TeTxLC-injected $Casp3^{+/+}$ mice, n=64 microglia from 5 Turq-injected $Casp3^{-/-}$ mice, and n=51 microglia from 6 TeTxLC-injected $Casp3^{-/-}$ mice.

The online version of this article includes the following figure supplement(s) for figure 6:

**Figure supplement 1.** Additional analysis on activity-dependent microglia-mediated engulfment of synapses.

To quantitatively detect engulfment of weak synapses by microglia, we compared TeTxLC-injected and control-injected animals of the same genotype (***Figure 6B, C and F***). We first calculated the ratio between the volume of right eye-originated RGC axon terminals and left eye-originated terminals that were engulfed by each microglia. Then, by comparing this right-to-left engulfment ratio in TeTxLC-injected vs. control-injected animals, engulfment of inactive synapses could be detected as higher engulfment ratios in TeTxLC-injected animals (***Figure 6F***). It is worth noting confounding factors, such as surgical manipulations, cancel out when calculating engulfment ratios, allowing information pertaining only to substrate preference to be isolated.

Consistent with previous studies (***Schafer et al., 2012***), in $Casp3^{+/+}$ mice inactivation of retinogeniculate synapses from the right eye led to enhanced microglia-mediated engulfment of right eye-originated RGC axon terminals but not left eye-originated terminals (***Figure 6—figure supplement 1A–B***), resulting in a significant increase in right-to-left engulfment ratios in microglia of TeTxLC-injected animals compared to control-injected animals (***Figure 6B, C and F***). In contrast, engulfment ratios were similar between TeTxLC-injected and control-injected $Casp3^{-/-}$ mice (***Figure 6D,E and G***, ***Figure 6—figure supplement 1C-D***), suggesting that microglia could no longer distinguish between strong and weak synapses in the absence of caspase-3 activation. These observations remain valid when data were averaged and reported by animal (***Figure 6—figure supplement 1E–F***). Summing the evidence obtained thus far, we propose the activity-dependent caspase-3 activation is a crucial signaling event that integrates upstream information about synaptic activity, takes place at weak/inactive synapses, and leads to activity-dependent elimination of weak synapses by microglia.

## Caspase-3 deficiency protects against amyloid-β-induced synapse loss

Lastly, we wanted to test if caspase-3 regulates synapse loss in neurodegenerative diseases through mechanisms analogous to its role in activity-dependent synapse elimination. To this end, we focused on Alzheimer's disease (AD). AD is the most common cause of dementia (*Knopman et al., 2021*), and synapse loss is the strongest correlate of cognitive decline in AD patients (*Meftah and Gan, 2023*). Accumulation and deposition of oligomeric or fibril amyloid-β (Aβ) peptide has been proposed to be the cause of neurodegeneration in AD (*Knopman et al., 2021*). Intriguingly, oligomeric Aβ impairs long-term potentiation (LTP) (*Shankar et al., 2008*), and this suppression of LTP requires caspase-3 activation (*Jo et al., 2011*). Additionally, caspase-3 activation has been linked to early synaptic dysfunction in an AD mouse model (*D'Amelio et al., 2011*).

To test if synapse loss in AD is regulated by caspase-3, we utilized the APP/PS1 mouse line, in which amyloid precursor protein (APP) and presenilin 1 (PS1) carrying mutations associated with early-onset familial AD are overexpressed in CNS neurons (*Jankowsky et al., 2004*). We observed that female *App/Ps1$^{+/-}$* mice developed amyloid deposits in the hippocampus and cortex by 5 mo (*Figure 7—figure supplement 1Aii*), and that the amyloid burden worsened by 6 mo (*Figure 7—figure supplement 1Aiii*). In male *App/Ps1$^{+/-}$* mice, amyloid deposits only became apparent at 6 mo and occurred at a lower level compared to age-matched females (*Figure 7—figure supplement 1Aiv-v*). We focused on analyzing *App/Ps1$^{+/-}$* female mice in subsequent experiments to obtain robust phenotypes. To quantify synapse loss, we labeled presynaptic and postsynaptic compartments in the dentate gyrus of 6-mo-old *App/Ps1$^{-/-}$* and *App/Ps1$^{+/-}$* littermates by detecting synaptic vesicle protein 2 (SV2) and Homer1, respectively, and selected multiple fields of interest per animal for imaging analysis while avoiding regions adjacent to amyloid deposits (*Figure 7—figure supplement 2*). We fitted ellipsoids to SV2 and Homer1 signals to identify pre- and post-synaptic puncta and defined synapses as Homer1-SV2 puncta pairs that were less than 300 nm apart (*Figure 7A*). We observed that synapse density in *App/Ps1$^{+/-}$* mice was significantly reduced compared to that in *App/Ps1$^{-/-}$* littermate controls (*Figure 7A–B*), and that this reduction could be attributed to lower densities of both pre- and post-synaptic puncta in *App/Ps1$^{+/-}$* mice (*Figure 7—figure supplement 3A–B*). We then crossed APP/PS1 mice to caspase-3 deficient mice and quantified synapse density in 6-mo-old *Casp3$^{-/-}$; App/Ps1$^{-/-}$* and *Casp3$^{-/-}$; App/Ps1$^{+/-}$* littermate mice (*Figure 7—figure supplement 2*). We observed that, in the caspase-3 deficient background, overexpression of mutant APP and PS1 no longer resulted in a reduction in the density of synapses (*Figure 7C–D*) or pre-/post-synaptic puncta (*Figure 7—figure supplement 3C–D*), suggesting that caspase-3 activity is required for Aβ-induced synapse loss in the APP/PS1 mouse model. Intriguingly, levels of amyloid deposition were comparable between *App/Ps1$^{+/-}$* and *Casp3$^{-/-}$; App/Ps1$^{+/-}$* mice (*Figure 7—figure supplement 1Aiv and B*), and reactive microglia were observed near amyloid deposits regardless of caspase-3 deficiency (*Figure 7—figure supplement 4*), indicating that protection conferred by caspase-3 deficiency against Aβ-induced synapse loss might not require inhibition of amyloid accumulation or microgliosis. We also investigated whether Aβ accumulation induced elevated levels of caspase-3 activity. We analyzed 4, 5, and 6-mo-old *App/Ps1$^{-/-}$* and *App/Ps1$^{+/-}$* mice but only observed robust upregulation of caspase-3 activity in the dentate gyrus of a subset of 4-mo-old *App/Ps1$^{+/-}$* mice (*Figure 7—figure supplement 5A–C*). Upregulated caspase-3 activity in these mice remained in a punctate pattern without inducing neuronal death (*Figure 7—figure supplement 5D–F*). It is possible that Aβ transiently induces caspase-3 activation in *App/Ps1$^{+/-}$* mice, but stochasticity in the onset and duration of such induction prevents reliable detection with our sample size and temporal resolution. Overall, our results demonstrate caspase-3 as an important regulator of Aβ-induced synapse loss and a potential therapeutic target for AD treatment.

# Discussion

## A model for caspase-3-dependent synapse elimination

Our findings support a model where competitive interaction between strong synapses and weak synapses in the retinogeniculate pathway triggers caspase-3 activation in the postsynaptic compartments of weak synapses. Caspase-3 activation results in the removal of weaker synaptic connections by microglia, thereby facilitating the establishment of mature circuits in the visual pathway.

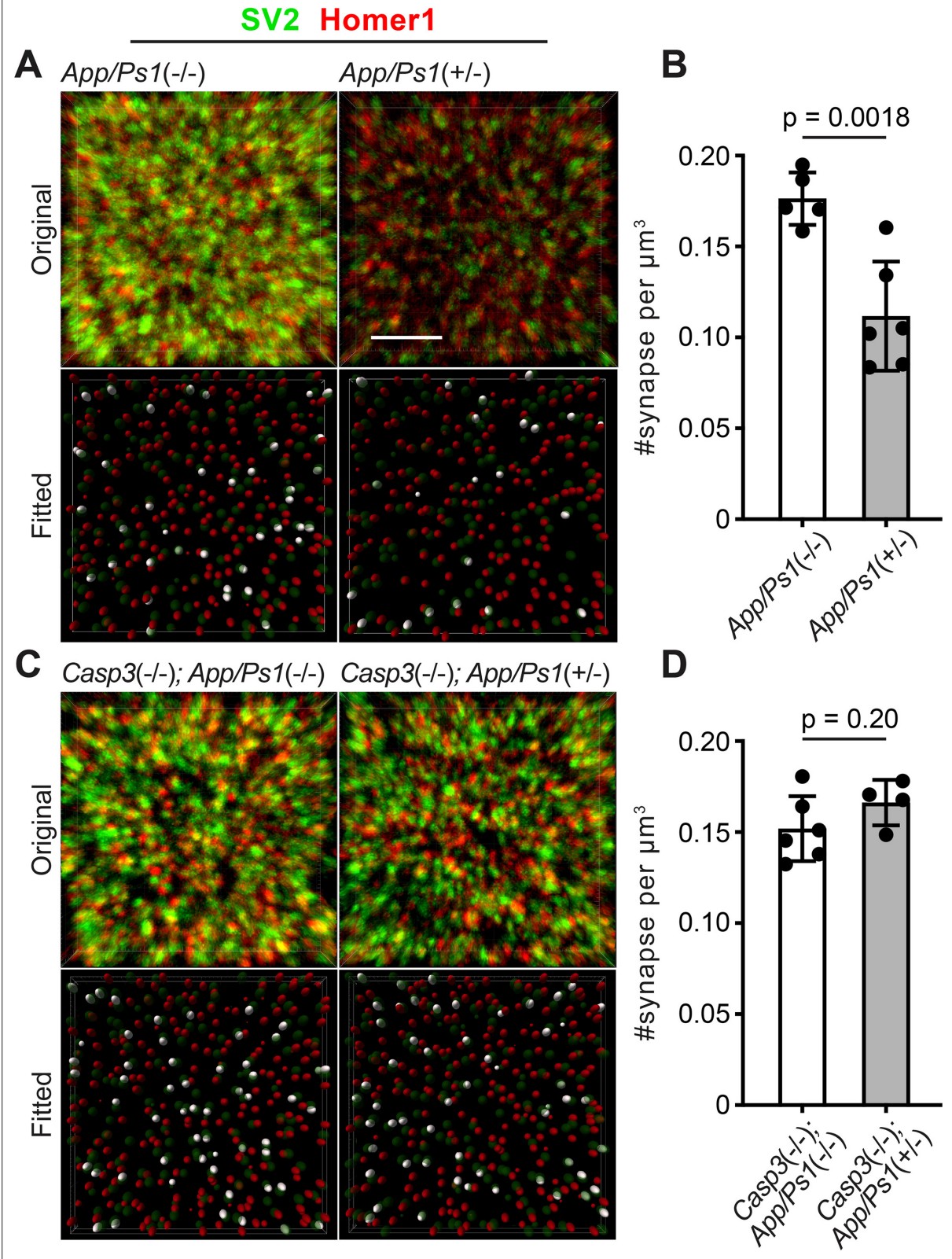

**Figure 7.** Caspase-3 deficiency protects against Aβ-induced synapse loss. (**A and C**) Representative 3D-reconstructed images (3 µm in z) showing presynaptic (SV2, in green) and postsynaptic (Homer1, in red) signals in dentate gyrus of female 6-mo-old amyloid precursor protein (APP)/PS1 mice on caspase-3 wild-type (**A**) and deficient (**C**) backgrounds. Ellipsoids were fitted to the original images (upper panels) to isolate pre- (green) and post-synaptic (red) puncta (lower panels). Homer1 ellipsoids found with 300 nm of a SV2 ellipsoid are highlighted in white in the fitted images. Original

*Figure 7 continued on next page*

*Figure 7 continued*

images are adjusted to the same contrast. For the fitted images, only ellipsoids from the upper half of the z-stack are shown. Scale bar represents 4 µm. (**B and D**) Quantification of synapse density in APP/PS1 mice on caspase-3 wild-type (**B**) and deficient (**D**) backgrounds. Mean and S.D. are shown. p-values were calculated from unpaired two-tailed t-tests. n=5 for *App/Ps1^-/-* mice, n=6 for *App/Ps1^+/-* mice, n=6 for *Casp3^-/-*; *App/Ps1^-/-* mice, and n=4 for *Casp3^-/-*; *App/Ps1^+/-* mice.

The online version of this article includes the following figure supplement(s) for figure 7:

**Figure supplement 1.** Amyloid deposition amyloid precursor protein (APP)/PS1 mouse lines.

**Figure supplement 2.** Selection of field of interest for synapse loss analysis.

**Figure supplement 3.** Additional analysis of synapse density in amyloid precursor protein (APP)/PS1 mouse lines.

**Figure supplement 4.** Microgliosis in amyloid precursor protein (APP)/PS1 mouse lines.

**Figure supplement 5.** Aβ-induced caspase-3 activity in amyloid precursor protein (APP)/PS1 mice.

## Mechanisms of activity-dependent caspase-3 activation

In the experiment where we demonstrated activity-dependent caspase-3 activation, we inactivated all retinogeniculate synapses from one eye to induce sufficiently high caspase-3 activity that can be discerned from background. Widespread caspase-3 activity induced by such strong inactivation at multiple sites in a given cell likely self-amplifies through positive feedback (*McComb et al., 2019*) and overwhelms negative regulation mechanisms (e.g. ubiquitin-dependent proteasomal degradation *Ertürk et al., 2014*; *Choi et al., 2009*; *Suzuki et al., 2001*), leading to the observed apoptosis of entire dLGN relay neurons (*Figure 1B*). However, apoptotic neurons were infrequently observed in the dLGN of control animals (*Figures 1D and 2B*), and caspase-3 deficiency does not significantly alter dLGN relay neuron numbers (*Figure 3—figure supplement 2A–B*), suggesting that during normal development, activity-dependent caspase-3 activation is predominantly localized and generally does not lead to neuronal death.

Intriguingly, synaptic inactivation in our experiment was achieved through TeTxLC expression in presynaptic RGCs, but caspase-3 activation was observed in postsynaptic relay neurons. Conversely, while caspase-3 is activated in the postsynaptic compartment, we were able to detect defects in synapse elimination using assays that measure presynaptic inputs. These observations suggest that activity-dependent caspase-3 activation requires synaptic transmission, and that caspase-3-dependent synapse elimination is not strictly axonal or dendritic but involves both pre- and post-synaptic compartments. An important limitation of this study is that experiments in full-body *Casp3^-/-* mice cannot query the roles of caspase-3 in presynaptic and postsynaptic compartments separately. Future experiments that perturb caspase-3 activation and other signaling events specifically in the pre- or post-synaptic compartment are needed to reveal more detailed mechanisms of activity-dependent synapse elimination.

The observation that activity-dependent caspase-3 activation requires the presence of both weak and strong synapses resonates with previous research demonstrating that relative synaptic efficacy biases the outcome of synapse elimination during development of neuromuscular junctions and the callosal pathway (*Yasuda et al., 2021a*; *Nagappan-Chettiar et al., 2023*; *Buffelli et al., 2003*; *Lee, 2020*). However, the mechanisms linking synaptic competition with caspase-3 activation remain elusive. During LTD, caspase-3 is activated through mitochondrial release of cytochrome-c (*Li et al., 2010*). Whether synaptic competition could induce caspase-3 activation via mechanisms such as calcium signaling, which regulates both neurotransmission and cytochrome-c release (*Garrido et al., 2006*), is an outstanding question.

## Link between caspase-3 activation and synapse engulfment by microglia

How does caspase-3 activation at weak synapses lead to synapse engulfment by microglia? Caspase-3 activation is known to trigger the display of apoptosis markers that can be recognized by phagocytes (*Park and Kim, 2017*). One of the best understood markers is phosphatidylserine (PtdSer), a phospholipid that translocates from the cytoplasmic leaflet of the plasma membrane to the exoplasmic leaflet during apoptosis (*Park and Kim, 2017*). Importantly, caspase-3 directly regulates the enzymes that translocate PtdSer through enzymatic cleavage (*Suzuki et al., 2013*; *Segawa et al., 2014*). Two recent studies demonstrated that PtdSer is displayed on retinogeniculate synapses during postnatal

development, and microglia use the exposed PtdSer as a cue to target and engulf synapses (*Li et al., 2020*; *Scott-Hewitt et al., 2020*). Whether PtdSer exposure is upregulated in response to synapse weakening and whether caspase-3 activity is required for such exposure are important questions that require further investigation.

The complement protein C1q has been observed to bind to PtdSer on apoptotic cells and facilitate phagocytic clearance (*Park and Kim, 2017*). Additionally, a previous study found caspase-3 activity in C1q-positive synaptosomes (*Győrffy et al., 2018*), suggesting that C1q may bridge caspase-3 activation and synapse elimination. However, we investigated C1q expression and localization in the dLGN of P5 TeTxLC-injected mice but did not observe upregulation of C1q abundance or colocalization between C1q and caspase-3 activity (data not shown).

It is often assumed in the literature that microglia directly engulf intact synapses, but existing data also support an alternative model where microglia indirectly contribute to synapse elimination by engulfing synaptic debris that is shed through neuron-autonomous mechanisms (*Eyo and Molofsky, 2023*). The observation that synaptic activity promotes microglial engulfment through caspase-3 activation raises the possibility that weakened synapses partially disintegrate through apoptosis-like mechanisms prior to being engulfed by microglia. Further studies are needed to elucidate the intermediate steps between caspase-3 activation and microglial engulfment of synapses.

## Caspase-3 as a potential target for AD treatment

Our findings suggest that caspase-3 deficiency confers protection against Aβ-induced synapse loss in a mouse model of AD. Notably, the absence of caspase-3 does not seem to affect Aβ deposition or microglia reactivity, indicating that caspase-3 depletion may mitigate synapse loss by directly countering Aβ-induced synaptic toxicity. Given the prominence of Aβ and inflammation as targets for AD drug development (*van Bokhoven et al., 2021*), inhibiting caspase-3 activity offers a potential alternative therapeutic strategy that could complement and enhance current treatment approaches. Therefore, it is important to test whether pharmacological inhibition of caspase-3 can effectively slow synapse loss and cognitive decline in both mouse models of AD and human patients.

# Materials and methods

## Mouse lines

The *Casp3^{-/-}* mouse line was a generous gift from Dr. Richard Flavell. This mouse line was received as cryopreserved sperm and was rederived using C57Bl/6 J oocytes. The resulting colony was maintained by continuous backcrossing to C57Bl/6 J mice. We verified that mice in our *Casp3^{-/-}* colony were congenic to the C57Bl/6 J background (greater than 99% identity by SNP analysis). It is important to note that regular backcrossing is necessary to maintain phenotype stability in *Casp3^{-/-}* mice. A small fraction of *Casp3^{-/-}* mice displayed varying degrees of hydrocephalus. These animals were excluded from our analyses. *Cx3cr1^{Gfp}* mice (stock number 005582) were obtained from the Jackson Laboratory. *Aldh1l1-Gfp* transgenic mice (stock number 011015-UCD) were obtained from Mutant Mouse Resource and Research Centers (MMRRC) as cryopreserved sperm and rederived using C57Bl/6 J oocytes. *App/Ps1* mice were obtained from the Jackson Laboratory (stock number 034832). Mice carrying *App/Ps1* transgenes were given DietGel 76 A (ClearH$_2$O, 72-07-5022) and were euthanized at the age of 6 mo if not used in experiments. For experiments associated with *Figures 1 and 2*, C57Bl/6NCrl mice were obtained from Charles River Laboratories (strain code 027). For experiments associated with *Figure 6*, C57Bl/6 J mice were obtained from the Jackson Laboratory (stock number 000664). All animal procedures were approved by IACUC and documented in protocol #21–0208.10 titled 'Studying neuronal cell biology and glia-neuron interactions in rodent models'.

## Reagents

Reagents were obtained from the following sources: isoflurane liquid (NDC 50989-150-15) was from Vedco; phosphate buffered saline without calcium and magnesium (PBS$^-$, BP2438) was from Fisher Scientific; bicuculline (0130) and cyclothiazide (0713) were from Tocris Bioscience; PBS with calcium and magnesium (PBS$^+$, SH30264) was from Cytiva; dimethyl sulfoxide (DMSO, D2650), bovine serum albumin (BSA, A9647), triton X-100 (T8787), Thioflavin S (T1892) were from Millipore-Sigma; paraformaldehyde powder (19200) was from Electron Microscopy Science; RPMI 1640 medium (11875093),

DMEM medium (11965092), fetal bovine serum (FBS, 16000044), penicillin/streptomycin (pen/strep, 15140122), CD Hybridoma medium (11279023), GlutaMAX supplement (35050061), protein A/G agarose beads (20421), protein A/G IgG binding buffer (54200), IgG elution buffer (21004), 1 M Tris-HCl pH 8.0 (15568025), 10% normal goat serum (50062Z), ProLong Gold mounting medium (P36934), Alexa Fluor 546 succinimidyl ester (A20002), cholera toxin subunit beta (CTB) conjugated to AlexaFluor (AF) 488, AF555, AF594 and AF647 (C22841, C22842, C22843, C34778), goat anti-rabbit secondary antibodies conjugated to AF488plus (A32731), AF555plus (A32732), AF594plus (A32740), and AF647plus (A32733), goat anti-chicken secondary antibody conjugated to AF488plus (A32931), goat anti-guinea pig secondary antibody conjugated to AF488 (A11073) and AF594 (A11076) were from ThermoFisher Scientific; anti-cleaved caspase-3 antibody (9661) was from Cell Signaling Technologies; anti-GFP antibody (GFP-1010) was from Aves Lab; anti-MAP2 antibody (NB300-213) was from Novus Biologicals; anti-NeuN antibody (266004) was from Synaptic Systems; anti-RBPMS antibody (1832-RBPMS) was from PhosphoSolutions; anti-Iba1 (ab178846) and anti-Homer1 (Ab97593) antibodies were from Abcam; anti-Cd68 antibody (MCA1957) was from Bio-Rad; hybridoma line producing anti-SV2 antibody (clone name SV2) was from DSHB.

## DNA cloning and AAV preparation

To make the pAAV-hSyn-mTurquoise2 plasmid (deposited as Addgene #239206), we ordered the DNA fragment for mTurquoise2 (*64*) as a Gblock from IDT and cloned it via isothermal assembly into a linearized AAV expression plasmid (derived from pAAV-Syn-iCre; Addgene #122518) under the control of human synapsin promoter. To make the pAAV-hSyn-TeTxLC-P2A-mTurquoise2 plasmid (deposited as Addgene #239210), the tetanus toxin light chain (TeTxLC) sequence along with the P2A sequence (2 A peptide from porcine teschovirus-1 polyprotein) at the 3' end was obtained via PCR from Addgene plasmid #159102 and fused in-frame with the mTurquoise2 DNA sequence in the same plasmid backbone as the pAAV-hSyn-mTurquoise2 plasmid. Note that in the main text, we used pAAV-hSyn-TeTxLC to refer to pAAV-hSyn-TeTxLC-P2A-mTurquoise2 because AAV carrying this plasmid induced very low levels of mTurquoise2 expression in vivo. When we needed endogenous fluorescence labeling, we chose to premix AAV carrying pAAV-hSyn-TeTxLC-P2A-mTurquoise2 with other AAVs expressing fluorescent proteins (at 1:1 ratio).

The pAAV-hSyn-tdTomato plasmid was cloned and provided by Janelia Viral Tools and has been deposited as Addgene #239048. The pAAV-CAG-GFP plasmid was previously made by Svoboda lab and is available as Addgene #28014.

The recombinant AAVs were prepared by Viral Tools of Janelia Core Services. Briefly, 2 d before transfection, 293T cells were seeded in three 150 mm T/C dishes at $1\times10^7$ cells/dish in DMEM medium supplemented with 10% FBS and cultured at 37 °C with 5% $CO_2$. The cells were transfected with 84 µg of pAAV plasmid, DJ capsid plasmid (*65*), and pHelper plasmid (Agilent 240071) at a ratio of 5:2:3 in polyethylenimine. After 6–8 hr, the cells were replenished with serum-free DMEM and further incubated for 3 d. The recombinant AAVs were collected from both the cells and supernatant and purified by two rounds of continuous cesium chloride density gradient centrifugation, dialyzed, concentrated, and filter sterilized. Titer (genome copies per ml) was determined by digital PCR with primers targeting the ITR. For all in vivo experiments a titer of at least $10^{13}$ genome copies/ml was used.

## Intraocular injections

In utero intraocular injections were performed as previously described (*Yasuda et al., 2021b*) with modifications. Briefly, pregnant mouse females with embryos at E15 gestation were anesthetized by 3% isoflurane inhalation, abdomens shaved, eye lube applied to each eye, then transferred to a nose cone and set to maintenance level of 2% isoflurane inhalation anesthesia. Warm fluids (lactated Ringer's solution + 5% dextrose, Strategic Applications Inc), Buprenorphine (0.1 mg/kg, Butler Animal Health), and Ketoprofen (5 mg/kg, Covetrus) were administered subcutaneously just prior to starting surgery. The animal was placed in dorsal recumbency, and the abdomen was cleansed using alternating swabs of 70% ethanol and 4% Chlorhexidine Gluconate (Molnlycke Health Care). A sterile drape was placed over the abdomen and using aseptic technique, the abdominal skin and wall were cut at midline, before gently removing the uterus from the abdomen. While keeping the uterus/embryos hydrated with warm saline sterile solution, the embryos were gently positioned with ring forceps to hold each embryo securely for injection. A previously filled glass pipette (Drummond Scientific), beveled to

30–40 µm at the opening was utilized to deliver constructs utilizing a pre-programmed Nanoject III microinjector (Drummond Scientific). 200–250 nl of AAVs prep was injected through the uterine wall and into each eye at a speed of 50 nl per second. To confirm proper targeting within the eye, the AAV solution was pre-mixed with 0.05% Fast Green (Sigma Aldrich). After all injections were completed, the uterus/embryos were carefully placed back into the abdomen and the abdomen was flushed with warm sterile saline. The abdominal wall and skin were sutured, and Marcaine (Hospira) was applied subcutaneously below the skin incision. A small drop of Vetbond (3 M) was applied to seal the skin incision. The animal was then removed from the anesthesia apparatus and allowed to recover on a heating pad before being placed back into a clean cage.

For intraocular injection at P4 and P9, pups with desired genotypes were anesthetized in an induction chamber filled with 2.5–3% isoflurane using a calibrated vaporizer. After the onset of anesthesia, isoflurane is reduced to 1.25%, and the mouse was removed from the chamber and placed under a dissection microscope. Proparacaine ophthalmic solution (0.5%, Sandoz, NDC 61314-016-01) was applied to each eye to be injected. A re-breathing nose cone was used to deliver 1.25–2% isoflurane throughout the procedure. A gentle blunt dissection was performed at each eyelid junction utilizing closed No. 5 Dumont forceps (Fine Science Tools) to create a 2–3 mm opening. A 33-gauge sharp beveled needle attached to a Hamilton microliter syringe (Hamilton Company) was used to draw up CTB-AF solutions. Just prior to injecting, the eye was proptosed and held securely in place by the Dumont forceps in preparation for injection. While holding the eye securely in place, the sclera was punctured with the Hamilton needle at the level of the ora serrata to inject 2.5 µL (for P9) or 2 µL (for P4) of desired CTB-AF solution into each eye. After the injection, eye lube was applied, the eye was allowed to move back into the socket, and the eyelids were gently pressed close to protect the eye. Pups were allowed to recover from anesthesia on a heated pad and put back into the cage with the dam.

## Monoclonal antibody production and labeling

SV2 hybridoma cells were thawed and seeded in RPMI 1640 medium containing 10% FBS and 1% Pen/Strep at 0.3 million cells/ml and cultured in an incubator at 37°C with 5% $CO_2$. After reaching $1\times10^6$ viable cells/ml density, the cells were diluted 1:2 by adding an equal volume of serum-free CD Hybridoma medium supplemented with 4% GlutaMax$^{Tm}$ and returned to the incubator. This dilution process was repeated two more times so that the cells reached $1\times10^6$ viable cells/ml density in a mixed medium containing ~1% FBS. Cell viability obtained was at least 90%. The cells were then spun down at 150×G for 10 min and resuspended in complete CD Hybridoma medium at $5\times10^5$ cells/ml density. The cells were further expanded by adding more complete CD Hybridoma medium when cells reach $1\times10^6$ cells/ml density. Once the desired total volume is reached (typically 100 ml), cells were cultured for 2 wk without medium change.

Culture supernatant containing SV2 antibody was cleared of hybridoma cells by centrifuging at 1000×G for 10 min. Cleared supernatant was diluted 1:1 with IgG binding buffer and passed through 1 ml of protein A/G agarose bead slurry (Thermo Fisher 20421, corresponding to 500 µl of settled resin) packed in a gravity flow column. The column was washed with IgG binding buffer, and SV2 antibody was eluted in five 1 ml fractions using IgG elution buffer and immediately neutralized with 10% volume of 1 M Tris-HCl, pH 8.0. Fractions with highest antibody concentration were pooled, desalted with Zeba columns (Thermo Fisher 89891), and concentrated to ~1 mg/ml using Amicon centrifuge filters (Millipore UFC801024). After adding glycerol (25% final), antibodies were flash-frozen in liquid nitrogen and kept at –80°C.

To label the primary SV2 antibodies, we used Alexa Fluor 546 succinimidyl ester. Briefly, we buffer-exchanged and concentrated SV2 IgG to 3 mg/ml using Amicon filters (UFC501024 Millipore Sigma) in phosphate-buffered saline that was brought to pH 8.3 with 100 mM $NaHCO_3$. Dimethyl sulfoxide-dissolved Alexa Fluor 546 succinimidyl ester was gently mixed with the SV2 IgG at 10:1 molar ratio and incubated for 2 hr at room temperature on a rotating shaker. Alexa Fluor 546 labeled IgG were purified from unreacted succinimidyl esters with the aid of PD MiniTrap Sephadex G-25 column (Cytiva 28918007) in PBS$^-$ by gravity flow following the manufacturer's instructions. The degree of labeling was determined by spectrophotometry on Nanodrop One (Thermo Fisher) by accounting for the Alexa Fluor 546 extinction coefficient at 554 nm and the corrected protein absorbance at 280 nm. Typically, a degree of labeling of 8–9 mols of dye per mol of protein was obtained. Labeled SV2 were

aliquoted and flash-frozen in 25% glycerol for long-term storage at –80°C at a final concentration of approximately 0.75 mg/ml.

## Immunohistochemistry

The mice were deeply anesthetized in a chamber filled with saturating isoflurane gas and transcardially perfused first with PBS⁻ and then with paraformaldehyde (PFA) solution (4% w/v in 20.3 mM $NaH_2PO_4$ and 79.7 mM $Na_2HPO_4$, pH = 7.4). The brains were removed and post-fixed in 4% PFA solution overnight at 4°C with gentle shaking. Fixed brains were washed 3×15 min in PBS⁻ at room temperature and stored in PBS⁻ at 4°C. For long-term storage, fixed brains were cryoprotected in 30% Sucrose (w/v in PBS⁻) for 72 hr and frozen in Tissue-Tek O.C.T. compound (Sakura) and stored at –80°C.

Brains stored in PBS⁻ were embedded in 4% agarose, and coronal sections (50 µm thick for mice younger than P10 or 40 µm thick for adult mice) were prepared using a vibratome (Leica VT 1200 S). For brains frozen in O.C.T. compound, coronal sections were prepared using a cryostat (Leica 3050 S). Sections were floated in 24-well plates in PBS⁻ and stored at 4°C. For antibody staining, free-floating sections were first permeabilized/blocked in 10% normal goat serum containing 0.3% (w/v) triton X-100 for 1 hr. The sections were then incubated with primary antibodies diluted in staining buffer (PBS⁺ containing 2% BSA w/v and 0.3% triton X-100) at 4°C overnight with gentle agitation. Dilutions of primary antibodies were: anti-cleaved caspase-3, 1:500; anti-GFP, 1:2000; anti-MAP2, 1:1000; anti-NeuN, 1:500; anti-Iba1, 1:500; anti-Cd68, 1:500; anti-SV2, dilute to 2.5 µg/ml; anti-Homer1, 1:400. The sections were then washed 3×15 min in washing buffer (PBS⁺ containing 0.1% triton X-100) and stained with appropriate fluorophore-conjugated secondary antibodies diluted in staining buffer for 3 hr at room temperature with gentle agitation. Secondary antibodies were always used at 1:500 dilution. The sections were washed 3x15 min in washing buffer, rinsed 3x5 min in PBS⁺, and mounted onto glass slides (Fisher, 12-550-15). After air-drying completely, mounted sections were immersed in ProLong Gold and coverslipped (Fisher, 12-541-024). The sections were imaged after 24 hr of curing at room temperature. The sections were protected from light during the entire process.

To prepare whole-mount retinae, eyes were removed from PFA-perfused mice and placed in PBS⁻. The cornea was punctured with a #11 scalpel, and several anterior-to-posterior cuts were made on the sclera starting from the puncture. Intact retinae were isolated by tearing and peeling off the sclera along the cuts. 4–5 radial cuts were made on each retina to flatten the tissue. Isolated retinae were post-fixed for 15 min in 4% PFA solution at room temperature with gentle agitation and washed for 3×5 min in PBS⁻ at room temperature with gentle agitation. Free-floating retinae were then stained with primary (anti-RBPMS antibody, used at 1:200 dilution) and secondary (goat anti-guinea pig Alexa Fluor 594 plus) antibodies and mounted onto glass slides as described above.

For staining of amyloid-β (Aβ) plaques, a 1% Thioflavin S stock solution (w/v, dissolved in ddH₂O) was diluted 100-fold in 50% ethanol (prepared by mixing pure ethanol and ddH₂O at 1:1 ratio) to create a 0.01% (w/v) working solution. The stock and working solutions were discarded after each use. Free-floating brain sections were dehydrated by washing 2×5 min in 50% ethanol, stained in 0.01% Thioflavin S solution for 8 min, washed for 2×5 min in 50% ethanol, then re-hydrated by washing 3x5 min in PBS⁺. All staining steps were performed at room temperature with gentle agitation. The sections were protected from light throughout the procedure. Stained sections were then mounted and cured as described above.

## Measuring synapse inactivation-induced caspase-3 activity

For experiments associated with *Figure 1*, *Figure 1—figure supplement 4*, E15 C57Bl/6NCrl mice were injected in the right eyes either with AAV carrying hSyn-mTurquoise2 (control group), or with a 1:1 mixture of AAV carrying hSyn-mTurquoise2 and AAV carrying hSyn-TeTxLC-P2A-mTurquoise2 (synapse inactivation group). At the age of P5, brains of both groups of mice were harvested and sectioned as described above. To quantify caspase-3 activity in the dLGN and the localization of activated caspase-3 relative to TeTxLC-expressing RGC axons (*Figure 1*), sections were stained with anti-cleaved caspase-3 antibody (using goat anti-rabbit AF594plus as the secondary) and anti-GFP antibody (using goat anti-chicken AF488plus as the secondary). Alternatively, sections were co-stained with anti-cleaved caspase-3 (using goat anti-rabbit AF647plus as the secondary) and anti-MAP2 antibody (using goat anti-rabbit AF488plus as the secondary) (*Figure 1—figure supplement 4A*), or with anti-cleaved caspase-3 (using goat anti-rabbit AF594plus as the secondary) and anti-NeuN antibody

(using goat anti-rabbit AF488plus as the secondary) (*Figure 1—figure supplement 4C*). Immunostaining was performed as described above.

For experiments associated with *Figure 2*, E15 C57Bl/6NCrl mice were injected in the left eyes with AAV carrying hSyn-tdTomato and in the right eyes with AAV carrying CAG-eGFP (control group), or in the right eyes with AAV carrying hSyn-mTurquoise2 and AAV carrying hSyn-TeTxLC-P2A-mTurquoise2 (single inactivation group), or in the left eyes with AAV carrying hSyn-tdTomato and AAV carrying hSyn-TeTxLC-P2A-mTurquoise2 and in the right eyes with AAV carrying CAG-eGFP and AAV carrying hSyn-TeTxLC-P2A-mTurquoise2 (dual inactivation group). At the age of P5, brains of all three groups of animals were harvested, sectioned, and stained with anti-cleaved caspase-3 antibody and goat antirabbit AF647plus secondary antibodies as described above.

To quantify caspase-3 activity in the dLGN (*Figure 1B–E*), confocal images of dLGNs were acquired on a Leica Stellaris 8 microscope using a 10 x/0.4 NA dry objective (Leica HC PL Apo CS2 10 x/0.4 dry). The channels were acquired sequentially (line-by-line) with the anti-caspase-3 first and anti-mTurquoise2 second. A 590 nm laser and emission read of 596 nm–748 nm was used for the anti-cleaved caspase-3 channel, and 440 nm and 488 nm lasers and an emission read of 457 nm–579 nm were used for anti-mTurquoise2 channel. The imaging parameters (laser power, pixel dwell time, signal averaging or accumulation) were set to maximize signal-to-background values while avoiding detector (SiPM type Power HyD S) saturation. Photon detection was set in counting mode and acquired as 12-bit images at 227 nm×227 nm pixel size. All datasets used for measurement of caspase-3 activity were acquired as Z-stacks (seven slices spaced 2.4 µm apart). To measure the total amount of active caspase-3 signal in the 3D stacks of dLGN we used the Labkit (*66*) Fiji plugin to generate and train a pixel classifier through Imaris v. 10 (Oxford Instruments). The same pixel classifier and surface creation settings were applied to data from both the control and the synapse inactivation groups to generate surface objects of active caspase-3 signal. All active caspase-3 objects found within the dLGN area (manually delineated based on mTurquoise2 signal) were selected, and the sum of voxel intensities the active caspase-3 objects was calculated and divided by the area of dLGN. The final values plotted for each mouse are averages of two consecutive sections.

To image active caspase-3 signals and TeTxLC-expressing RGC axons at high resolution (*Figure 1G*), we used a Leica Stellaris 8 confocal microscope and a 63 x/1.4 NA oil objective (Leica HC PL APO 63 x/1,40 OIL CS2) in 'Lightning' mode (in LasX 4.6) set at high resolution grade, which reduces the pinhole size to 0.5 Airy units. Laser and emission settings were identical to those used the previous section that described quantification of active caspase-3 signal in mTurquoise2-labeled dLGNs. 3D image stacks (31.5 µm×31.5 µm×12 µm) were acquired at high voxel density (43 nm×43 nm×204 nm) in photon counting mode and 4 x signal accumulation. Subsequently, the images were processed by iterative deconvolution using a theoretical point spread function informed by the imaging parameters of the microscope and constrained by an 'Adaptive' strategy that accounts for local background and signal-to-noise ratio information.

Confocal images of MAP2 and active caspase-3 signals in dLGN (*Figure 1—figure supplement 4A*) were similarly imaged on a Leica Stellaris 8 microscope with a 63 x/1.4 NA oil objective (Leica HC PL APO 63 x/1,40 OIL CS2). A 653 nm laser line and an emission read of 662–829 nm were used for caspase-3 channel, and a 499 nm laser line and an emission read of 504–549 nm were used for Map2 channel. The channels were acquired simultaneously at voxel densities of 65 nm×65 nm× 299 nm. NeuN and active caspase-3 signals in dLGN (*Figure 1—figure supplement 4C*) were imaged using a 20 x/0.7 NA water objective (Leica HC PL APO 20 x/0,7 IMM CORR CS2). A 590 nm laser and an emission read of 595–750 nm were used for imaging caspase-3 and a 499 nm laser with a 509–595 nm emission read was used for the NeuN signal. The channels were acquired sequentially (active caspase-3 first) at pixel densities of 142 nm×142 nm.

## Measuring overlap of eye-specific territories in dLGN

For comparison between Casp3[+/+] and Casp3[-/-] animals (*Figure 3*), Casp3[+/-] males were bred to Casp3[+/-] females to generate Casp3[+/+] and Casp3[-/-] littermate pups. Pups were genotyped at birth. Before intraocular injection, lyophilized CTB-AF488 and CTB-AF594 powders were reconstituted in 1% DMSO in PBS[-] to a final concentration of 5 mg/ml. CTB solutions were aliquoted and stored at –20 °C for later use. Thawed CTB solutions were discarded after each surgery. Intraocular injection of CTB-AF solutions was performed at the age of P9 as described above. CTB-AF488 was injected into

left eyes and CTB-AF594 into right eyes. 24 hr after the injection, brains of P10 pups were harvested, and 50 µm thick coronal sections prepared as described above.

Cured sections were imaged on a Zeiss AxioObserver Z.1 microscope equipped with an LSM 880 Airy scan detector and an EC Plan-Neofluar 10 x objective (NA = 0.3). A 488 nm laser and a 499–544 nm detection window were used for CTB-AlexaFlour488, and a 594 nm laser and a 597–659 nm detection window were used for CTB-AlexaFlour594. Signals from the two channels were acquired sequentially into 16-bit images with a 0.4 µm×0.4 µm pixel size. Laser power and detector gain settings were maintained at the same level for most images, with occasional adjustment to avoid saturation. One image per dLGN per section was acquired.

To quantify the overlap between contralateral-specific and ipsilateral-specific territories in the dLGN, we selected 7–8 sections per animal that span the majority of left dLGN and analyzed their images in ImageJ. For each image, a mask for the entire dLGN region and a mask for a background region in the thalamus with no labeling were manually created. Background signal from each channel was estimated as the average signal in the background region and was subtracted from signals in the dGLN region. Then the background-subtracted signals in the dLGN region were normalized to between 0 and 1 for each image and each channel. We picked 7 increasing cutoff thresholds (0.1, 0.125, 0.15, 0.175, 0.2, 0.225, and 0.25), and for each threshold and each channel, signals that are greater than or equal to the threshold were considered real signals. For each animal (corresponding to 7–8 images of left dLGN) and each threshold, we calculated percentage overlap as the ratio between the total number of pixels with real signals in both channels in all images and the total number of pixels in the entire dLGN region of all images. We plotted the percentage overlap as a measurement of segregation of eye-specific territories in dLGN. Similar analyses were performed on right dGLN images yielding similar results.

For overlap analysis presented in *Figure 1—figure supplement 3*, E15 mouse embryos were injected in their right eyes either with AAV carrying hSyn-mTurquoise2 (control group) or a 1:1 mixture of AAV carrying hSyn-TeTxLC-P2A-mTurquoise2 and AAV carrying hSyn-mTurquoise2 (inactivation group). The left eyes of both groups of mice were injected with AAV carrying the hSyn-tdTomato. At P8 (when segregation of eye territories is largely complete and, thus sufficient for us to validate the effect of TeTxLC on synapse inactivation), brains from both groups were harvested, and 50 µm thick coronal sections were prepared as described above. Only animals that had well-labeled dLGNs on both sides were selected for analysis. Mounted and cured sections containing dLGN were imaged in confocal mode on a Leica Stellaris 8 microscope using a 10 x/0.4 NA dry objective (Leica HC PL Apo CS2 10 x/0.4 dry). A 441 nm laser and a 445–548 nm detection window were used for the mTurquoise2 channel, and a 554 nm laser and a 564–721 nm detection window were used for tdTomato channel. The images were acquired at 8-bit depth with a 0.227 µm×0.227 µm pixel size (at 1 Airy unit for l=580 nm) and at a laser power, pixel dwell time, and signal accumulation sufficient to avoid detector (Power HyD S in counting mode) saturation. One image per dLGN per section was acquired. For each animal, overlap was measured as described above on the left dLGN (the side where mTurquoise2 signal occupied the majority of the dLGN) by averaging from 3 to 5 consecutive sections. The cutoff thresholds used were: 0.050, 0.075, 0.1, 0.125, 0.15, 0.175, 0.2.

For overlap analysis presented in *Figure 2—figure supplement 1*, embryos in the control group were injected at E15 with AAV carrying tdTomato construct in the left eyes, and with AAV carrying the CAG-GFP construct in the right eyes. Embryos in the dual inactivation group were injected at E15 with a mixture (1:1) of AAV carrying hSyn-tdTomato construct and AAV carrying hSyn-TeTxLC-P2A-mTurquoise2 construct in their left eyes, and with a mixture (1:1) of AAV carrying the CAG-GFP construct and AAV carrying the hSyn-TeTxLC-P2A-mTurquoise2 construct in their right eyes. At P10, the brains were harvested and sectioned as described above. Confocal images of dLGN were acquired on a Leica Stellaris 8 microscope using a 10 x/0.4 NA dry objective (Leica HC PL Apo CS2 10 x/0.4 dry). A 488 nm laser line with a 488 nm notch filter and a 485–559 nm detection window were used for the GFP channel, and a 554 nm laser and a 559–701 nm detection window were used for the tdTomato channel in sequential line mode to avoid any cross-excitation (with tdTomato channel acquired first). The images were acquired at 12-bit depth with a 0.248 µm×0.248 µm pixel size (at 1 Airy unit for l=580 nm) and at a laser power, pixel dwell time, and signal accumulation that avoided detector (Power HyD S in counting mode) saturation. One image per dLGN per section was acquired. For each

animal, overlap was measured as described above on the right dLGN (the side where tdTomato signal occupied the majority of the dLGN) by averaging from five consecutive sections.

## Quantification of RGC densities in the retina

To estimate the density of RGCs in the retina (*Figure 3—figure supplement 1*), retinae of P10 *Casp3*[+/+] and *Casp3*[-/-] mice were harvested and stained with anti-RBPMS antibody as described above. We imaged four large square regions of interest (1.16 mm×1.16 mm) in each retina on four sides of the optic disc. The images were acquired with a Leica TCS SP8 Laser Scanning Microscope using the 10 x/0.4 NA dry objective (Leica HC PL Apo CS2 10 x/0.4 dry), a 594 nm laser line, and an emission window of 607–694 nm, at a pixel sampling of 302 nm×302 nm. To account for tissue unevenness, we collected a stack of three Z-slices (2.4 µm apart). To count RGCs, we sampled three subregions of a size of 100 µm$^2$ from each large region of interest. The subregions were selected to be at similar distances from the optic disc and in areas devoid of tissue tears. Cells were modeled to spots by fitting them to ellipsoid objects (XY size = 8 µm; Z size = 16 µm, with background subtraction) in Imaris 9.6 (Oxford Instruments). Densities of spots for each region were averaged and reported as number of RGC per 100 µm$^2$ of retina.

## Quantification of dLGN neuron densities

To determine the density of relay neurons in dLGN (*Figure 3—figure supplement 2*), we collected and stained 50 µm brain sections with anti-NeuN antibody as described above. Confocal image stacks (Leica Stellaris 8) were acquired with a 10 x/0.4 NA dry objective (Leica HC PL Apo CS2 10 x/0.4 dry) at a voxel sampling of 210 nm x 210 nm x 2410 nm. For DAPI (i.e. cell nuclei) signal, the 405 nm laser with 430–550 nm emission was used, while for the NeuN signal, revealed by Alexa594 conjugated IgG, the 590 nm laser with 597–750 nm emission was used. For each animal, two consecutive coronal histological sections were imaged as Z stacks containing eight Z-slices that were centered on dLGN. Volumes of NeuN signal were segmented in Imaris 10.2 (Oxford Instruments). To improve the identification of neuronal cell bodies, we generated synthetic images from the geometric means of DAPI and segmented NeuN images and applied a spot object finder algorithm (6 µm diameter) to the entire stack. Finally, we selected the spots corresponding to dLGN only and reported the number of spots over the area of dLGN (manually traced for each histological section). Each data point in *Figure 3—figure supplement 2* is an average of dLGN densities from two brain sections of the same animal.

## Whole-cell patch-clamp recording in acute brain slices

To characterize electrophysiological properties of retinogeniculate synapses in *Casp3*[+/+] and *Casp3*[-/-] mice (*Figure 4*), the mice (aged P28 – P32) were decapitated under deep isoflurane anesthesia, and the brain was removed and quickly transferred to an ice-cold dissection solution containing (in mM): 194 sucrose, 30 NaCl, 2.5 KCl, 1.2 NaH$_2$PO$_4$, 26 NaHCO$_3$, 10 D-Glucose, 1 MgCl$_2$ (pH 7.4, oxygenated with 95% CO$_2$ and 5% O$_2$). Parasagittal brain slices containing the optic tract and dLGN were prepared using a vibratome (Leica VT 1200 S) as previously described (*Chen and Regehr, 2000*). The slices were recovered at 35.5 °C in an incubation chamber (BSC-PC, Warner Instrument, USA) filled with artificial cerebrospinal fluid (ACSF) containing (in mM): 124 NaCl, 2.5 KCl, 1.2 NaH$_2$PO4, 26 NaHCO$_3$, 10 D-Glucose, 2 CaCl$_2$, 1 MgCl$_2$ (pH 7.4, oxygenated with 95% CO$_2$ and 5% O$_2$, osmolarity ~310). After ~30 min of recovery, the chamber was maintained at room temperature.

All recordings were performed with a MultiClamp 700B amplifier (Molecular Devices) and signals were filtered at 2 kHz and digitized at 20 kHz with via USB-6343 (National Instruments) under the control of WaveSurfer software (https://wavesurfer.janelia.org). All recordings were carried on slices submerged in the recording chamber of an upright microscope (BX61WI; Olympus, Tokyo, Japan) equipped with IR-DIC (infrared-differential interference contrast) microscopy and a water-immersion objective lens (60 X, 1.00 NA; Olympus). Slices were maintained under continuous perfusion of oxygenated ACSF at 29–31°C. Patch pipettes were pulled from thin-wall single-barrel borosilicate glass (TW150-6, WPI), resulting in electric resistance of 3~5 MΩ when filled with an intracellular solution containing the following (in mM): 120 cesium methane sulfonate, 5 NaCl, 10 tetraethylammonium chloride, 10 HEPES, 4 lidocaine N-ethyl bromide, 1.1 EGTA, 4 magnesium ATP, and 0.3 sodium GTP, with pH adjusted to 7.2 with CsOH and osmolality set to ~290 mOsm.

Whole-cell voltage-clamp recordings of relay neurons in the dLGN were performed as previously described (*Figure 4A–C*; *Chen and Regehr, 2000*). The slices were submerged in ACSF containing 20 µM bicuculline to inhibit GABA$_A$ receptors, and a bipolar concentric stimulation electrode (FHC) was placed on the surface of the optic tract to deliver stimuli with intensities ranging from 0 to 100 µA (200 µs duration) with an inter-trial interval of 40–60 s. In the whole-cell configuration (series resistance <20 MΩ), the membrane potential of the neuron was clamped at –70 mV to measure evoked excitatory postsynaptic current (EPSC) mediated by activation of AMPA receptors, and then at +40 mV to measure evoked EPSC mediated by activation of NMDA receptors. The stimulus intensity was systemically increased until a response to the stimulus was observed, and then, was reduced to a previous level where no response was detected. From that level, the stimulus intensity was increased by 0.5 µA steps to recruit single inputs which were putative single fiber (SF) amplitudes. When the EPSC amplitude reached a plateau, it was considered a putative maximum response. If the EPSC amplitude dropped while the stimulation intensity increased, data from that relay neuron was discarded.

To measure miniature excitatory postsynaptic current (mEPSC) (*Figure 4D–F*), the slices submerged in the ACSF containing TTX 1 µM and bicuculine 20 µM to inhibit spontaneous action potentials and GABA$_A$ receptors. The membrane potential was clamped at –70 mV and synaptic current was recorded for 180 s. After the recording, a –9 pA threshold was used to detect mEPSC.

To measure paired-pulse ratio (PPR) (*Figure 4G–H*), the slices were submerged in the ACSF containing 20 µM bicuculline and 50 µM cyclothiazide to inhibit GABA$_A$ receptors and AMPAR desensitization, and the stimulation electrode was placed on the optic tract. In the whole-cell configuration, the membrane potential was clamped at –70 mV. After confirming the stimulation intensity that evoked the maximum response, two consecutive electrical stimuli of that intensity (300 µs duration) were delivered at 50, 150, 250, 500, and 1000 ms inter-stimulus intervals. Each pair of stimuli constitutes one trial, and two consecutive trials were separated by an interval of 40 s. For each inter-stimulus interval, two trials were performed and averaged. PPR was calculated as the maximum amplitude of the second EPSC over the maximum amplitude of the first EPSC. If in any trial except for the first trial the first EPSC dropped significantly in amplitude compared to the first EPSC of the first trial, data from that dLGN relay neuron was discarded.

## Measuring microglia- and astrocyte-mediated synapse engulfment in vivo

To measure microglia-mediated synapse engulfment in vivo (*Figure 5*, *Figure 5—figure supplement 2*), we bred *Casp3$^{-/-}$* males to *Casp3$^{+/-}$*; *Cx3ct1-Gfp$^{+/-}$* females to generate *Casp3$^{-/-}$*; *Cx3ct1-Gfp$^{+/-}$* and *Casp3$^{+/-}$*; *Cx3ct1-Gfp$^{+/-}$* littermate pups. To measure astrocyte-mediated synapse engulfment in vivo (*Figure 5—figure supplement 3*), we bred *Casp3$^{-/-}$* males to *Casp3$^{+/-}$*; *Aldh1l1-Gfp$^{+/-}$* to generate *Casp3$^{-/-}$*; *Aldh1l1-Gfp$^{+/-}$* and *Casp3$^{+/-}$*; *Aldh1l1-Gfp$^{+/-}$* littermate pups. P4 pups with desired genotypes were injected with 2 µL of CTB-AF555 in left eyes and 2 µL of CTB-AF647 in right eyes as described above. 24 hr later, the brains were harvested, sectioned, and mounted as described above.

To analyze engulfment of synapses, the sections were imaged using a Leica TCS SP8 Laser Scanning Microscope equipped with an HC Plan Apo CS2 40 x (NA = 1.3) oil objective and a White Light Laser Unit. The 3D multichannel stacks with a voxel size of 0.075×0.075×0.312 µm for microglia samples or 0.048×0.048×0.2 µm for astrocyte samples were acquired as 16-bit images with 4 x line averages for microglia or 2 x for astrocytes at 400 Hz scanner speed. The stacks were set to collect a volume of 20 micrometers in the Z dimension and 387.5 micrometers in XY dimensions for microglia or 288 µm in XY for astrocytes. The pinhole was set at 1 Airy unit for 520 nm wavelength. The excitation lasers and detection windows were set to minimize crosstalk and bleed-through as well as the saturation of detectors while simultaneously imaging the three channels. A 478 nm laser and a detection window of 488–537 nm were used for the GFP channel, a 550 nm laser and a detection window of 596–629 nm were used for CTB-AF555, and a 651 nm laser and a detection window of 665–750 nm were used for CTB-AF647. Imaris 9.6 (Oxford Instruments) was used to segment microglia, astrocytes, ipsilateral RGC axon terminals, and contralateral RGC terminals from the 3D stacks using the Surface Creation module with background subtraction. For segmentation of RGC axon terminals and astrocytes, all parameters used were left at default values (auto). For segmentation of microglia, we set the threshold manually at the highest inflection curve in the histogram of voxel intensities, which in

our experience provided a good trade-off for good segmentation of cell bodies and major processes while keeping the adjustment consistent across different image datasets. A background subtraction ball size of 1 µm was used for microglia, a ball size of 1.4 µm was used for astrocytes, and a ball size of 0.5 µm was used for AF555 and AF647 labeled axonal materials. The segmented signal of AF555 and AF647 was each converted to a binary mask of 0 and 1 voxels. Finally, the sum of the segmented volume (i.e. sum of ones) of AF555 and/or AF647 in each microglia or astrocyte was either directly plotted or normalized to the microglia or astrocyte volume and plotted. To control the inherent signal variability across LGN, only microglia and astrocytes present in the ipsilateral region of the 3 middle sections of the left dLGN of each mouse were included in the analysis. Also, only microglia or astrocytes whose soma and proximal processes were contained in the volumetric stack were included in the final analysis. All littermate pups of the same genotype were pooled together.

## Measuring activity-dependent engulfment of synapses by microglia

To investigate the role of caspase-3 in activity-dependent synapse elimination (*Figure 6*, *Figure 6— figure supplement 1*), we used C57Bl/6 J wild-type mice as controls and bred *Casp3*$^{-/-}$ males with *Casp3*$^{+/-}$ females to generate *Casp3*$^{-/-}$ mice. Note that C57Bl/6 J and *Casp3*$^{-/-}$ mice used in this experiment are not littermates and should not be directly compared with each other. C57Bl/6 J and *Casp3*$^{-/-}$ embryos were injected at E15 in their right eye either with AAV carrying hSyn-TeTxLC-P2A-mTurquoise2 or AAV carrying hSyn-mTurquoise2 as described above. To label the RGC axons, at P4, the pups were injected with 2 µL of CTB-AF555 in left eyes and 2 µL of CTB-AF647 in right eyes as described above. At P5, the brains were harvested and sectioned. Two 50 µm sections that spanned the central part of the dLGN were stained with anti-Iba1 antibody and goat anti-rabbit AF488 secondary (as described above) to label microglial cell bodies.

Mounted and cured sections were then imaged on Leica Stellaris 8 confocal microscope using a 63 x/1.4 NA oil objective (Leica HC PL APO 63 x/1,40 OIL CS2). Four stitched 3D-stacks were acquired for each left dLGN (the side where CTF-AF647 and mTurquoise2 occupied the majority of the dLGN) to generate a 3D volume of 366 µm×366 µm in XY and approximately 20 µm in the Z dimension, with a voxel size of 64 nm×64 nm×299 nm. The channels were acquired sequentially: first using a 499 nm laser with an emission window of 507–556 nm together with a 653 nm laser with an emission window of 663–750 nm for Iba1-AF488 and CTB-AF647 signal, respectively, then using a 553 nm laser with an emission window of 558–663 nm for the CTB-AF555 signal. Images acquired were collected as 12-bit datasets, and care was taken to set the laser power, signal accumulation, and pixel dwell time to avoid detector saturation. Only animals whose dLGN areas were fully labeled with both AF555 and AF647 as well as the mTurquoise signals were imaged and analyzed. Animals in which CTB signal appeared widespread throughout the brain parenchyma were excluded from the analysis as these labeling patterns were likely due to surgical mistargeting or issues with tissue integrity.

To segment volumes corresponding to microglia and CTB-AF555 and CTB-AF647 labeled RGC axon terminals, we utilized the Surface Creation module with machine learning segmentation of Imaris 10.1.1 (Oxford Instruments). We first trained for several rounds a pixel classifier for each channel using two independent image files and saved the classifiers along with the other settings (smoothing grain size of 100 nm and discarding volumes smaller than 10 voxels) as 'Favorite Creation Parameter.' Subsequently, the same Favorite Creation Parameters were applied to all data sets, with the rare instances when microglia segmentation was slightly tuned via the same pixel classifier to better capture some of the cell border areas. Similar to the previous analyses (*Figure 5*), the segmented signals of CTB-AF647 and CTB-AF555 were converted to binary images (0 or 1 voxels), and the total volume of CTB-AF647- or CTB-AF555-positive voxels in each microglia was extracted, converted to fraction of the microglia volume, and plotted. Only the cells whose body and major processes were bound within the 3D stack and present within the middle area of dLGN (where the ipsilateral signal is present) were included in the analysis. Data from the same genotype and treatment were pooled together. Either one or two sections were analyzed for each animal.

## Measuring microglia activation

To measure if lack of caspase 3 causes microglia activation (*Figure 5—figure supplement 1*), we bred *Casp3*$^{+/-}$ males with *Casp3*$^{+/-}$ females to generate *Casp3*$^{-/-}$ and Casp3$^{+/+}$ littermates. After genotyping, at P5, the brains were harvested and sectioned coronally as described above. Two 50 µm sections that

spanned the central part of the dLGN were stained with anti-Iba1 antibody (detected by goat anti-rabbit AF488 secondary) to label microglial cell bodies and anti-CD68 antibody (detected by donkey anti-rat AF555 secondary) to detect CD68 protein. Mounted and cured sections were imaged on Leica Stellaris 8 confocal microscope using a 63 x/1.4 NA oil objective (Leica HC PL APO 63 x/1,40 OIL CS2). A 3D image stack of 185 μm x 185 μm in XY and approximately 20 μm in the Z dimension, with a voxel size of 72 nm×72 nm×299 nm was acquired for each dLGN. The channels were acquired in photon counting mode simultaneously: a 499 nm laser with an emission window of 504–548 nm together with a 553 nm laser with an emission window of 560–650 nm for Iba1-AF488 and CD68-AF555. To eliminate any fluorescence emission bleed-through between channels, the photon arrival times were gated to 3–5 ns for Iba1-AF488 and to 0–1.5 ns for Iba1-AF555 channels, respectively. Images acquired were collected as 8-bit datasets, and care was taken to set the laser power, signal accumulation, and pixel dwell time to avoid fluorophore bleaching or detector saturation.

For segmentation of microglia and CD68 signals, we utilized the Surface Creation module with machine learning segmentation of Imaris 10.1.1 (Oxford Instruments). We first trained for several rounds a pixel classifier for each channel then saved the classifiers along with the other settings (smoothing grain size of 145 nm and discarding volumes smaller than 10 voxels) as 'Favorite Creation Parameter.' Subsequently, the same Favorite Creation Parameters were applied to all data sets, with the rare instances when microglia segmentation was slightly adjusted via the same pixel classfier to better capture some of the cell border areas. Similar to the previous analyses (*Figure 5*), the segmented signals of CD68-AF555 were converted to binary images (0 or 1 voxels), and the total volume CTB-AF555-positive voxels in each microglia was extracted, converted to fraction of the microglia volume, and plotted. For measurement of relative CD68 intensity, the sum of CD68 signal was divided by the microglia volume. Only the microglia whose body and major processes were confined within the 3D stack were included in the analysis. All staining, imaging, and analysis parameters were identically applied to all sections. Two sections were analyzed for each animal. Data from the same genotype and were pooled together.

## Characterization of *App/Ps1* transgenic mice

To detect Aβ deposition (*Figure 7—figure supplement 1*), brains of 6-mo-old female *App/Ps1-/-* mice, 5-mo-old and 6-mo-old male and female *App/Ps1+/-* mice, and 6-mo-old female *Casp3-/-; App/Ps1+/-* mice were harvested, sectioned, and stained with Thioflavin S as described above. Stained sections were imaged on a Leica Stellaris 8 microscope using a 10 x/0.4 NA dry objective (Leica HC PL Apo CS2 10 x/0.4 dry). A 460 nm laser line was used for excitation and a window of 469–600 nm was used for emission. Tile images (pixel size: 284 nm×284 nm) were collected and merged into one image spanning an entire coronal section in Las X 4.6 Navigator. To quantify the level of amyloid deposition in 6-mo-old female *App/Ps1+/-* and *Casp3-/-; App/Ps1+/-* mice, the number of Thioflavin S-positive plaques was manually counted in a total of 3 sections per animal and averaged.

To investigate the role of caspase-3 in Aβ-induced synapse loss (*Figure 7*), *Casp3-/-* mice were bred to *App/Ps1+/-* mice to create *Casp3-/-; App/Ps1-/-* and *Casp3-/-; App/Ps1+/-* mice. Brains of 6-mo-old female *App/Ps1-/-* and *App/Ps1+/-* littermates as well as *Casp3-/-; App/Ps1-/-* and *Casp3-/-; App/Ps1+/-* littermates were harvested, sectioned, and stained with anti-SV2 antibody (conjugated to AF546) and anti-Homer1 antibody (using goat anti-rabbit AF647 as the secondary) to label presynaptic and postsynaptic compartments, respectively. Stained sections were imaged on a Leica Stellaris 8 confocal microscope using a 63 x/1.4 NA oil objective (Leica HC PL APO 63 x/1,40 OIL CS2). To acquire a representative dataset, we focused on the molecular layer of the dentate gyrus of the hippocampus from both the left and right side of the brain and sampled three regions from each side, as shown in *Figure 7—figure supplement 2*. 3D-stacks (61.5 μm×61.5 μm×20 μm) were acquired with a voxel size of 64 nm×64 nm×298 nm. 561 nm and 638 nm laser lines were used for excitation, 555–635 nm and 642–792 nm windows were used for emission reading, and appropriate notch filters were used for blocking the exciting laser lines. The channels were acquired as 8-bit data sets with detectors set in counting mode, and the lasers power, signal accumulation, and pixel dwell time set at levels that avoided detector saturation. To measure synapse number per unit volume, we first trimmed the raw 3D stacks to a stack of approximately 3.5 μm thickness on the side closest to the coverslip (to avoid any signal depletion in the middle part of the section due to antibody penetration issues) and cropped out regions where large gaps (corresponding to blood vessels) devoid of Sv2 or Homer1 staining

were present. The punctate signals of SV2 and Homer1 were fit in Imaris 10.1 (Oxford Instruments) to ellipsoids (500 nm in XY and 850 nm in Z for SV2; 400 nm in XY and 800 nm in Z for Homer1) with background subtraction ('Spot Creation'). Then we designated as 'synapses' all Homer1 ellipsoids that were within 300 nm of a SV2 ellipsoid (center-to-center distance). For each animal, the total number of synapses in all 3D-stacks analyzed was divided by the total volume of all stacks analyzed and plotted as a single value.

To detect Aβ-induced microglia activation (*Figure 7—figure supplement 4*), brains of 6-mo-old female *App/Ps1*[-/-], *App/Ps1*[+/-], and *Casp3*[-/-]; *App/Ps1*[+/-] mice were harvested, sectioned, and stained with anti-Iba1 antibody (using goat anti-rabbit AF647plus as the secondary) and anti-Cd68 (using goat anti-rat AF555plus as the secondary) as labels of microglia cell body and microglial activation, respectively. Stained sections were imaged on a Leica Stellaris 8 confocal microscope using a 20 x/0.7 NA water objective (Leica HC PL APO 20 x/0,7 IMM CORR CS2). Tile images spanning the hippocampus were acquired at voxel sizes of 142 nm×142 nm×300 nm. A 548 nm laser line and a 554–634 nm emission window were used for the CD68-AF555 channel, and a 653 laser line and a 663–748 nm emission window were used for the Iba1-AF647 channel. Microgliosis could be detected as clusters of microglia with stubby processes and strong intracellular Cd68 signals (*Figure 7—figure supplement 4*).

To quantify Aβ-induced caspase-3 activity (*Figure 7—figure supplement 5*), brains of 4-mo-old, 5-mo-old, and 6-mo-old female *App/Ps1*[-/-] and *App/Ps1*[+/-] littermates were harvested, sectioned, and stained with anti-cleaved caspase-3 antibody (using goat anti-rabbit AF555plus as the secondary). Stained sections were imaged on a Leica Stellaris 8 confocal microscope using a 63 x/1.4 NA oil objective (Leica HC PL APO 63 x/1,40 OIL CS2). We sampled at least 2 regions of interest from the molecular layer of the dentate gyrus of the hippocampus on each side of the brain. 3D-stacks (92.26 μm×92.26 μm×22 μm) were acquired with a voxel size of 81 nm×81 nm×299 nm. A 561 nm laser line was used for excitation and an emission window of 550–728 nm was used for the active caspase-3 channel. To measure the number of caspase 3 objects in the dataset, we first trimmed the 3D stacks to a volume where the signal is uniformly distributed (i.e. in focus) and segmented the objects using the 'Surface creation' analysis in Imaris 10.1 (Oxford Instruments). The dataset was first smoothed by a two-pixel filter, then a ball size of 400 nm was used to find the objects with background subtraction. Only objects bigger than 10 and smaller than 2000 voxels were kept. Finally, the number of caspase 3 objects was divided by the volume of the 3D stack analyzed and reported per brain side (left or right) for each animal.

To measure the relative density of granule cells in the dentate gyrus (*Figure 7—figure supplement 5*), brains of 6-mo-old female *App/Ps1*[-/-], *App/Ps1*[+/-], *Casp3*[-/-]; *App/Ps1*[-/-], and *Casp3*[-/-]; *App/Ps1*[+/-] mice were harvested, sectioned, and stained with anti-NeuN antibody (using goat anti-mouse AF594plus as the secondary) to label neuronal nuclei. Stained sections were imaged on a Leica Stellaris 8 confocal microscope using a 20 x/0.7 NA water objective (Leica HC PL APO 20 x/0,7 IMM CORR CS2). 3D stacks of the dentate gyrus (581 μm×581 μm×30 μm) were acquired at voxel sizes of 160 nm×160 nm×1000 nm for each section and side (left and right) using a 594 nm laser and an emission window of 597–750 nm. The channels were acquired as 8-bit images with the detector set in counting mode, and the lasers power, signal accumulation, and pixel dwell time set at levels that avoided detector saturation. To count the number of nuclei, images were first rotated to position vertically the cortical side of the dentate gyrus. A maximum projection of the brightest two Z-slices was extracted from each 3D stack, and NeuN + nuclei were manually counted within a rectangular region of interest (89 μm×150 μm) that was positioned on the granule cell layer at 300 μm away from the tip of the gyrus.

## Statistics

For all experiments, we aim to reach a sample size of n=7 mice for each experimental group. This sample size is determined based on comparing to similar prior studies and a statistical power calculation (performed with G*Power) that detects an effect size of 1.5–2 in an unpaired two-tailed t-test with $\alpha$=0.05 and power = 80% (yielding a sample size of 6–8).

In the case where comparisons between animals of different genotypes were made, animals were assigned to experimental groups according to their genotypes. In the case where we compared animals with the same genotype but received different treatments, animals were randomly assigned to experimental groups.

In the case where manual analysis of data is required (quantifying number of RGC innervations as shown *Figure 4C*), the persons performing the analysis were blinded to the genotypes of the animals. In cases where analyses were performed automatically with in-house written codes or commercial softwares (all other analyses in this paper), the persons performing the analysis were not blinded to the genotypes.

Animals were included in the study according to their age, genotype, and sex. Littermate controls were used whenever applicable. Animals were excluded based on pre-defined criteria: mice exhibit hydrocephalus (occurs infrequently in $Casp3^{-/-}$ mice), mice fail to recover from surgery, or surgery failure (overt tissue damage such as an deflated eye, lack of fluorescent labeling in the dLGN, widespread microglia activation in the dLGN, or widespread labeling throughout brain parenchyma). When immunohistochemistry was performed on tissue sections, sections with failed staining (for example, an abnormally high background signal when compared to its technical replicate) were excluded from analyses. In the case of microglia and astrocyte analyses in *Figure 5*, *Figure 5—figure supplement 3*, only cells in the region of interest were analyzed. If part of the cell body is not contained within the section, that cell is also excluded from the analyses. No exclusion of outlier datapoints were performed.

All replicates shown in this study are biological replicates. Experiments in this study were in general not repeated multiple times due to prohibitive costs and ethic constraints. To avoid possible batch effects, experiments were performed in mini-batches. For example, an experiment using seven wild-type and seven mutant animals were performed in three minibatches, containing 2, 2, and 3 animals of each genotype. The data from these mini-batches were then pooled and analyzed.

Statistical analyses were performed with GraphPad Prism. If sample distribution passed normality test, parametric tests (e.g. unpaired t-test, ANOVA with post-hoc Tukey test) were used. If normality was not observed, non-parametric tests (e.g. Mann-Whitney rank test) were used. The types of statistical tests used for each figure, their sample sizes, and their p-values have been reported in the figures and figure legends.

## Acknowledgements

We thank Crystall Lopez, Benjamin Gantz, and Brooke Groff for help with animal husbandry and breeding. We thank Sarah Lindo, Colin Morrow, Claire Boyer, and Rae Demars for their help with surgery. We thank Monique Copeland, Amy Hu, Morgan Clarke, and Benjamin Foster for their help with histology. We are also grateful to Sarah Kivimaki and the Viral Tools team of Janelia Core Services for help with AAV preparation. We thank Dr. Cagla Eroglu, Dr. Richard Axel, and Dr. David Clapham for their comments on the manuscript. This work is supported by the Howard Hughes Medical Institute.

## Additional information

### Competing interests

Erin K O'Shea: Member of eLife's Board of Directors and President of the Howard Hughes Medical Institute. The other authors declare that no competing interests exist.

### Funding

| Funder | Grant reference number | Author |
|---|---|---|
| Howard Hughes Medical Institute | | Zhou Yu<br>Andrian Gutu<br>Namsoo Kim<br>Erin K O'Shea |

The funders had no role in study design, data collection and interpretation, or the decision to submit the work for publication.

### Author contributions

Zhou Yu, Conceptualization, Resources, Data curation, Software, Formal analysis, Supervision, Validation, Investigation, Visualization, Methodology, Writing – original draft, Project administration, Writing

– review and editing; Andrian Gutu, Resources, Data curation, Software, Formal analysis, Validation, Investigation, Visualization, Methodology, Writing – review and editing; Namsoo Kim, Data curation, Software, Formal analysis, Investigation, Visualization, Methodology, Writing – review and editing; Erin K O'Shea, Conceptualization, Supervision, Funding acquisition, Project administration, Writing – review and editing

## Author ORCIDs
Zhou Yu ⓘ https://orcid.org/0000-0003-3213-4583
Andrian Gutu ⓘ http://orcid.org/0000-0002-5804-1978
Erin K O'Shea ⓘ https://orcid.org/0000-0002-2649-1018

Reviewer #2 (Public review): https://doi.org/10.7554/eLife.101779.3.sa1
Author response https://doi.org/10.7554/eLife.101779.3.sa2

---

# Additional files

## Supplementary files
MDAR checklist

Source data 1. CSV files containing raw quantification values plotted in all main figures and figure supplements.

Source code 1. Computer codes.

## Data availability
All data generated in this study are included in the manuscript and supporting files. Raw quantifications have been attached as *Source data 1*.

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
