## [Editor Report · eLife Assessment]

This study presents an **important** finding on the involvement of a Caspase 3-dependent pathway in the elimination of synapses for retinogeniculate circuit refinement and eye-specific territory segregation. This work fits well with the concept of "synaptosis" which has been proposed in the past. The evidence supporting the claims of the authors is **convincing**, demonstrating that caspase-3 activation is essential for microglial elimination of synapses during both brain development and neurodegeneration. The work will be of interest to investigators studying cell death pathways, neurodevelopment, and neurodegenerative disease.

---

## [Referee Report · Reviewer #2 (Public review)]

This manuscript by Yu et al. demonstrates that activation of caspase-3 is essential for synapse elimination by microglia, but not by astrocytes. This study also reveals that caspase-3 activation-mediated synapse elimination is required for retinogeniculate circuit refinement and eye-specific territories segregation in dLGN in an activity-dependent manner. Inhibition of synaptic activity increases caspase-3 activation and microglial phagocytosis, while caspase-3 deficiency blocks microglia-mediated synapse elimination and circuit refinement in the dLGN. The authors further demonstrate that caspase-3 activation mediates synapse loss in AD, loss of caspase-3 prevented synapse loss in AD mice. Overall, this study reveals that caspase-3 activation is an important mechanism underlying the selectivity of microglia-mediated synapse elimination during brain development and in neurodegenerative diseases.

A previous study (Gyorffy B. et al., PNSA 2018) has shown that caspase-3 signal correlates with C1q tagging of synapses (mostly using in vitro approaches), which suggests that caspase-3 would be an underlying mechanism of microglial selection of synapses for removal. The current study provides convincing in vivo evidence demonstrating that caspase-3 activation is essential for microglial elimination of synapses during both brain development and neurodegeneration.

---

## [Author Response]

The following is the authors’ response to the original reviews.

**Public Reviews:**

**Reviewer #1 (Public Review):**
In this manuscript, the authors study the effects of synaptic activity on the process of eye-specific segregation, focusing on the role of caspase 3, classically associated with apoptosis. The method for synaptic silencing is elegant and requires intrauterine injection of a tetanus toxin light chain into the eye. The authors report that this silencing leads to increased caspase 3 in the contralateral eye (Figure 1) and demonstrate evidence of punctate caspase 3 that does not overlap neuronal markers like map2. However, the quantifications showing increased caspase 3 in the silenced eye (done at P5) are complicated by overlap with the signal from entire dying cells in the thalamus. The authors also show that global caspase 3 deficiency impairs the process of eye-specific segregation and circuit refinement (Figures 3-4).

The reviewer states: “this silencing leads to increased caspase 3 in the contralateral eye”. We observed increased caspase-3 activity, not protein levels, in the contralateral dLGN, not eye.

The reviewer states: “and demonstrate evidence of punctate caspase 3 that does not overlap neuronal markers like map2”. We do not believe that this statement is accurate, as we show that the punctate active caspase-3 signals overlap with the dendritic marker MAP2 (Figure S4A).

The reviewer also states: “, the quantifications showing increased caspase 3 [*activity*] in the silenced [*dLGN*] (done at P5) are complicated by overlap with the signal from entire dying cells in the thalamus”. We do not believe that this statement is accurate. The apoptotic neurons we observed are relay neurons (confirmed by their morphology and positive staining of NeuN – Figure S4B-C) located in the dLGN (the dLGN is clearly labeled by expression of fluorescent proteins in RGCs, and only caspase-3 activity in the dLGN area is analyzed), not “cells” of unknown lineage (as suggested by the reviewer) in the general “thalamus” area (as suggested by the reviewer). If the dying cells were non-neuronal cells, that would indeed confound our quantification and conclusions, but that is not the case.

We argue that whole-cell caspase-3 activation in dLGN relay neurons is a *bona fide* response to synaptic silencing by TeTxLC and therefore should be included in the quantification. We have two sets of controls: one is between the strongly inactivated dLGN and the weakly inactivated dLGN in the same TeTxLC-injected animal; and the second is between the dLGN of TeTxLC-injected animals and mock-injected animals. In both controls, only the dLGNs receiving strong synapse inactivation have more apoptotic dLGN relay neurons, demonstrating that these cells occur because of synapse inactivation. It is also unlikely that our perturbation is causing cell death through a non-synaptic mechanism. Since mock injections do not cause apoptosis in dLGN neurons, this phenomenon is not related to surgical damage. TeTxLC is injected into the eyes and only expressed in presynaptic RGCs, not in postsynaptic relay neurons, so this phenomenon is also unlikely to be caused by TeTxLC-related toxicity. Furthermore, if apoptosis of dLGN relay neurons is not related to synapse inactivation, then when TeTxLC is injected into both eyes, one would expect to see either the same amount or more apoptotic relay neurons, but we instead observed a reduction in dLGN neuron apoptosis, suggesting that synapse-related mechanisms are responsible. Considering the above, occasional whole-cell caspase-3 activation in relay neurons in TeTxLC-inactivated dLGN is causally linked to synapse inactivation and should be included in the quantification.

We also revised the manuscript to better explain the possible mechanistic connection between localized caspase-3 activity and whole-cell caspase-3 activity. We propose that whole-cell caspase-3 activation occurs because of uncontrolled accumulation of localized caspase-3 activation. Please see line 127-140 and line 403-413 for details.

Additionally, we would like to clarify that we are not claiming that synapse inactivation leads to only localized caspase-3 activation or only whole-cell caspase-3 activation, as is suggested by the editors and reviewers in the eLife assessment. We have clearly stated in the manuscript that both types of signals were observed. However, we reasoned that, because whole-cell caspase-3 activation in unperturbed dLGNs – which undergo normal synapse elimination – is infrequently observed, whole-cell caspase-3 activation may not be a significant driver of synapse elimination during normal development. In this revision, we included a new experiment to corroborate this hypothesis. If whole-cell caspase-3 activation in dLGN relay neurons is a prevalent phenomenon during normal development, such caspase-3 activity would lead to significant death of dLGN relay neurons during normal development. Consequently, if we block caspase-3 activation by deleting caspase-3, the number of relay neurons in the dLGN should increase. However, in support of our hypothesis, we observed comparable numbers of relay neurons in *Casp3+/+* and *Casp3-/-* mice. Please see Figure S7 for details.

The authors also report that "synapse weakening-induced caspase-3 activation determines the specificity of synapse elimination mediated by microglia but not astrocytes" (abstract). They report that microglia engulf fewer RGC axon terminals in caspase 3 deficient animals (Figure 5), and that this preferentially occurs in silenced terminals, but this preferential effect is lost in caspase 3 knockouts. Based on this, the authors conclude that caspase 3 directs microglia to eliminate weaker synapses. However, a much simpler and critical experiment that the authors did not perform is to eliminate microglia and show that the caspase 3 dependent effects go away. Without this experiment, there is no reason to assume that microglia are directing synaptic elimination.

The reviewer states: “microglia engulf fewer RGC axon terminals in caspase 3 deficient animals (Figure 5), and that this preferentially occurs in silenced terminals, but this preferential effect is lost in caspase 3 knockouts”. We are not sure what the reviewer means by “this preferentially occurs in silenced terminals”. Our results show that microglia preferentially engulf silenced terminals, and such preference is lost in caspase-3 deficient mice (Figure 6).

We do not understand the experiment where the reviewer suggested to: “eliminate microglia and show that the caspase 3 dependent effects go away”. To quantify caspase-3 dependent engulfment of synaptic material by microglia or preferential engulfment of silenced terminals by microglia, microglia must be present in the tissue sample. If we eliminate microglia, neither of these measurements can be made. What could be measured if microglia are eliminated is the refinement of retinogeniculate pathway. This experiment would test whether microglia are required for caspase-3 dependent phenotypes. This is not a claim made in the manuscript. Instead, we claimed caspase-3 is required for microglia to engulf weak synapses, as supported by the evidence presented in Figure 6.

We did not claim that “microglia are directing synaptic elimination”. Our claim is that synapse inactivation induces caspase-3 activity, and caspase-3 activation in turn leads to engulfment of weak synapses by microglia. Based on this model, it is the neuronal activity that fundamentally directs synapse elimination. Synapse engulfment by microglia is only a readout we used to measure the outcome of activity-dependent synapse elimination. We have revised all sections in the manuscript that are related to synapse engulfment by microglia to emphasize the logic of this model.

We have also revised the abstract and title of the paper to better align it with our main claims, removed the reference to astrocytes, and clarified that microglia engulfment measurements are used as readouts of synapse elimination.

Finally, the authors also report that caspase 3 deficiency alters synapse loss in 6-month-old female APP/PS1 mice, but this is not really related to the rest of the paper.

We respectfully disagree that Figure 7 is not related to the rest of the paper. Many genes involved in postnatal synapse elimination, such as C1q and C3, have been implicated in neurodegeneration. It is therefore natural and important to ask whether the function of caspase-3 in regulating synaptic homeostasis extends to neurodegenerative diseases in adult animals. The answer to this question may have broad therapeutic impacts.

**Reviewer #2 (Public Review):**
Summary:This manuscript by Yu et al. demonstrates that activation of caspase-3 is essential for synapse elimination by microglia, but not by astrocytes. This study also reveals that caspase 3 activation-mediated synapse elimination is required for retinogeniculate circuit refinement and eye-specific territories segregation in dLGN in an activity-dependent manner. Inhibition of synaptic activity increases caspase-3 activation and microglial phagocytosis, while caspase-3 deficiency blocks microglia-mediated synapse elimination and circuit refinement in the dLGN. The authors further demonstrate that caspase-3 activation mediates synapse loss in AD, loss of caspase-3 prevented synapse loss in AD mice. Overall, this study reveals that caspase-3 activation is an important mechanism underlying the selectivity of microglia-mediated synapse elimination during brain development and in neurodegenerative diseases.Strengths:A previous study (Gyorffy B. et al., PNSA 2018) has shown that caspase-3 signal correlates with C1q tagging of synapses (mostly using in vitro approaches), which suggests that caspase-3 would be an underlying mechanism of microglial selection of synapses for removal. The current study provides direct in vivo evidence demonstrating that caspase-3 activation is essential for microglial elimination of synapses in both brain development and neurodegeneration.The paper is well-organized and easy to read. The schematic drawings are helpful for understanding the experimental designs and purposes.Weaknesses:It seems that astrocytes contain large amounts of engulfed materials from ipsilateral and contralateral axon terminals (Figure S11B) and that caspase-3 deficiency also decreased the volume of engulfed materials by astrocytes (Figures S11C, D). So the possibility that astrocyte-mediated synapse elimination contributes to circuit refinement in dLGN cannot be excluded.

We would like to clarify that we do not claim that astrocytes are unimportant for synapse elimination or circuit refinement. We acknowledge that the claim made in the original submitted manuscript that caspase-3 does not regulate synapse elimination by astrocytes lacks strong supporting evidence. We have removed this claim and revised the section related to synapse engulfment by astrocytes to provide a more rigorous interpretation of our data. We also removed the section in discussion regarding distinct substrate preferences of microglia and astrocytes.

Does blocking single or dual inactivation of synapse activity (using TeTxLC) increase microglial or astrocytic engulfment of synaptic materials (of one or both sides) in dLGN?

We assume that by “blocking single or dual inactivation of synapse activity”, the reviewer refers to inactivating retinogeniculate synapses from one or both eyes.

We showed that inactivating retinogeniculate synapses from one eye (single inactivation) increases engulfment of inactive synapses by microglia (Figure 6). We did not measure synapse engulfment by microglia while inactivating retinogeniculate synapses from both eyes (dual inactivation). However, based on the total active caspase-3 signal (Figure 2) in the dual inactivation scenario, we do not expect to see an increase in engulfment of synaptic material by microglia.

We did not measure astrocyte-mediated engulfment with single or dual inactivation, as we did not see a robust caspase-3 dependent phenotype in synapse engulfment by astrocytes.

**Recommendations for the authors:**

**Reviewer #1 (Recommendations for the Authors):**
(1) Figure 1 - It is not clear from this figure whether the authors are measuring caspase 3 in dendritic compartments or in dying relay neurons in the thalamus. The authors state that "either" whole cell death (1B) or smaller punctate signals (1F) were observed. When quantifying "photons" in Figure 1E, it appears most of the signal captured will be of dying relay neurons. What determined which signal was observed, and what is being quantified in Figure 1E? This also applies to the quantifications being reported in Figure 2.

The quantification includes both types of signals – it is sum of all active caspase-3 signal within the dLGN boundary. We note that there is a significant amount of punctate signal in the TeTxLC-inactivated dLGN. Unfortunately, due to file compression, these signals are not clearly visible in the submitted manuscript file. We have provided high resolution figures in this revision.

As argued above in the response to the public review, apoptotic relay neurons in TeTxLC-inactivated dLGN (not the general thalamus area) occur as a direct consequence of synapse inactivation. Therefore, active caspase-3 signals in these relay neurons should be included in the quantification.

We believe it is the extent of synapse inactivation (i.e., the number of synapses that are inactivated) that determines whether dLGN relay neuron apoptosis occurs or not. Such apoptosis is expected considering the nature of the apoptosis signaling cascade. In the intrinsic apoptosis pathway, release of cytochrome-c from mitochondria induces cleavage of the initiator caspase, caspase-9, and caspase-9 in turn cleaves the executioner caspases, caspase-3/7, which causes apoptosis. Caspase-3 can cleave upstream factors in the apoptosis pathway, leading to explosive amplification of caspase-3 activity (McComb et al., DOI: 10.1126/sciadv.aau9433). When a relay neuron receives a few inactivated synapses, caspase-3 activation in the postsynaptic dendrite can remain local (as we observed in Figure 1), constrained by mechanisms such as proteasomal degradation of cleaved caspase-3 (Erturk et al., DOI: 10.1523/JNEUROSCI.3121-13.2014). However, when a relay neuron receives many inactivated synapses, the cumulative caspase-3 activity induced in the dendrite can overwhelm negative regulation and lead to significantly higher levels of caspase-3 activity in entire dendrites (Figure S4B) through positive feedback amplification, eventually leading to caspase-3 activation in entire relay neurons. Please see line 127-140 and line 403-413 for our discussion in the main text.

(2) Figure 5 - Figures 5c-d and Fig 6 are confounded by pseudoreplication, whereby performing statistics on 50-60 microglia inflates statistical significance. Could the authors show all these data per mouse?

If we understand the reviewer correctly, the reviewer is suggesting that reporting measurements from multiple microglia in one animal constitutes pseudo-replication. This is correct in a strict sense, as microglia in the same animal are more likely to be similar than microglia from different animals. In the revised version, we have plotted the data by animal in Figure S11 and S13. The observations remain valid. However, we would like to point out that averaging measurements from all microglia in each animal and report by mouse is very conservative, as measurements from microglia in the same animal still vary greatly due to cell-to-cell differences.

(3) Although the authors are not the only ones to use this strategy, it is worth noting that performing all microglial experiments in Cx3cr1 heterozygotes could lead to alterations in microglial function that may not be reflective of their homeostatic roles.

We acknowledge that Cx3cr1 heterozygosity could cause alterations in microglial physiology.

While Cx3cr1 heterozygosity may impact microglia physiology, we note that the engulfment assay in Figure 5 is comparing microglia in *Cx3cr1+/-*; *Casp3+/-* and *Cx3cr1+/-*; *Casp3-/-* animals. Therefore, the impact of Cx3cr1 heterozygosity is controlled for in our experiment, and the observed difference in engulfed synaptic material in microglia is an effect specific to caspase-3 deficiency. However, we acknowledge that this difference could be quantitatively affected by Cx3cr1 heterozygosity.

It is important to note that we did not perform all microglia engulfment analyses using Cx3cr1^+/-^ mice. We have edited the manuscript to make this more clear. In the activity-dependent microglia engulfment analysis performed in Figure 6, we used *Casp3+/+* and *Casp3-/-* animals and detected microglia with anti-Iba1 immunostaining. Therefore, the impact of Cx3cr1 heterozygosity is not a problem for this experiment.

Minor:(1) Figures are presented out of order, which makes the manuscript difficult to follow.

We have revised text regarding the segregation analysis to align with the order of figures.

(2) Figure S3 is very confusing- the terms "left" and "right" are used in three or four partly overlapping contexts (which eye, which injection, which panel or subpanel of the figure is being referred to). Would this not be more appropriately analyzed with a repeated measures ANOVA (multiple comparisons not necessary) rather than multiple separate T-tests?

We have revised Figure S3 and S5 with better annotation and legends.

Yes, it is possible to use repeated measure two-way ANOVA. The analysis reports significant effect from genotypes, with a dF of 1, SoS and MoS of 0.0001081, F(1,13) = 7.595, and p = 0.0164. We used multiple separate t-tests because we wanted to show how genotype effects change with increasing thresholds, whereas two-way ANOVA only provides one overall p-value.

(3) Could the authors clarify why the percentage overlap (in the controls) is so different between Figure 3C and Figure S3C, and why different thresholds are applied?

This difference is primary due to difference in age. Figure 3 and Figure S5 are acquired at age of P10, while Figure S3 is acquired at P8. While the segregation process is largely complete by P8, the segregation continues from P8 to P10. Therefore, overlap measured at P10 will be lower than that measured at P8. If we compare overlap at the same threshold (e.g., 10%) and at the same age in Figure 3 and S5, the overlap is very similar.

The choice of threshold is related to the methods of labeling. In Figure 3, RGC terminals are labeled with AlexaFlour conjugated cholera toxin subunit-beta (CTB). In Figure S3 and S5, RGC axons are labeled by expression of fluorescent proteins. Labeling with CTB only labels membrane surfaces but yields stronger and slightly different signals at fine scales than labeling with fluorescent protein which are cell fillers. For Figure S3 and S5 (which use fluorescent protein labeling), higher thresholds such as those used in Figure 3 (which use CTB labeling) can be applied and the same trend still holds, but the data will be noisier. Regardless of the small difference in thresholds used, the important observation is that the defects in TeTxLC-injected or caspase-3 deficient animals are clear across multiple thresholds.

(4) Many describe the eye-specific segregation process as being complete "between P8-10". Other studies have quantified ESS at P10 (Stevens 2007). The authors state they did all quantifications at P8 (l. 82) and refer to Figure 3, but Figure 3 shows images from P10, whereas Figure S3 shows data from P8.

We did not say we performed all quantification at P8. In line 85, we said “To validate the efficacy of our synapse inactivation method, we injected AAV-hSyn-TeTxLC into the right eye of wildtype E15 embryos and analyzed the segregation of eye-specific territories at postnatal day 8 (P8), when the segregation process is largely complete”. The age of postnatal day 8 in this context is specifically referring to the experiment shown in Figure S3. For the segregation analysis in Figure 3, we specifically stated that the experiment was conducted at P10 (line 277).

Although the experiment in Figure S3 is conducted at P8, and Figure S5 and Figure 3 show results at P10, each dataset always included appropriate age-matched controls. P8 is generally considered an age where segregation is mostly complete and sufficient for us to assess the potency of TeTxLC-delivered AAV on eye segregation. We don’t think performing the experiment shown in Figure S3 at P8 impacts the interpretation of the data.

(5) Is Figure 6 also using Cx3cr1 GFP to label microglia? This is not clarified.

We apologize for this oversight. In Figure 6 microglia are labeled by anti-Iba1 immunostaining. We have clarified this in figure legends and text.

**Reviewer #2 (Recommendations for the Authors):**
(1) The authors quantified the caspase-3 activity using immunostaining and confocal microscopy (Figures 1B-E). They may need to verify the result (increased level of activated caspase-3 upon synapse inactivation) using alternative methods, such as western blotting.

Both western blot and immunostaining are based on antibody-antigen interaction. These two methods are not likely sufficiently independent. Additionally, to perform a western blot, we would need to surgically collect the TeTxLC-inactivated dLGN to avoid sample contamination from other brain regions. Such collection at the age we are interested in (P5) is very challenging. We have tested the anti-cleaved caspase-3 antibody using caspase-3 deficient mice and we can confirm it is a highly specific antibody that doesn’t generate signal in the caspase-3 deficient tissue samples.

(2) Does caspase-3 deficiency alter the density of microglia or astrocytes in dLGN?

No. Neither the density of microglia nor astrocytes changed with caspase-3 deficiency. In the case of microglia, we find that the mean density of microglia per unit area of dLGN is virtually the same in wild type and caspase-3 deficient mice (two-tailed t test P = 0.8556, 6 wild type and 5 Casp3^-/-^ mice). Some overviews showing microglia in dLGNs of wildtype and caspase-3 deficient mice can be found in Figure S10. Similarly for astrocytes, we did not observe overt changes in astrocytes dLGN density linked to caspase-3 deficiency.

(3) During dLGN eye-specific segregation in normal developing animals, did the authors observe different levels of activated caspase-3 in different regions (territories)?

For normal developing animals, the activated caspase-3 signal is generally sparse, and it is difficult to distinguish whether the signal is related to synapse elimination. For animals receiving TeTxLC-injection, we did notice that in the dLGN contralateral to the injection, where most inactivated synapses are located, the punctate caspase-3 signal tends to concentrate on the ventral-medial side of the dLGN (Figure 1B), which is the region preferentially innervated by the contralateral eye.

(4) Recording of NMDAR-mediated synaptic currents may not be necessary for demonstrating that caspase 3 is essential for dLGN circuit refinement. In addition, the PPR may not necessarily reflect the number of innervations that a dLGN neuron receives. Instead, showing the changes in the frequency of mEPSCs (or synapse/spine density) may be more supportive.

Thank you for the comment. We have performed the suggested mEPSC measurements and reported the results in revised Figure 4D-F.

(5) Why is caspase 3 activation enhanced (compared to control) only at 4 months of age, when A-beta deposition has not formed yet, but not at later time points in AD mice (Figure S17)?

A prevailing hypothesis in the field is that the form of A-beta that is most neurotoxic is the soluble oligomeric form, not the fibril form that leads to plaque deposition. As the oligomeric form appears before plaque deposition, the enhanced caspase-3 activation we observed at 4-month may reflect an increase in oligomeric A-beta, which occurs before any visible A-beta plaque formation.

(6) The manuscript can be made more concise, and the figures more organized.

We removed superfluous details and corrected text-figure mismatches in the revised manuscript to improve readability.